# How $\mu$-opioid receptor recognizes fentanyl

Quynh N. Vo [1,2], Paween Mahinthichaichan[1,2], Jana Shen [2✉] & Christopher R. Ellis [1✉]

Roughly half of the drug overdose-related deaths in the United States are related to synthetic opioids represented by fentanyl which is a potent agonist of mu-opioid receptor (mOR). In recent years, X-ray crystal structures of mOR in complex with morphine derivatives have been determined; however, structural basis of mOR activation by fentanyl-like opioids remains lacking. Exploiting the X-ray structure of BU72-bound mOR and several molecular simulation techniques, we elucidated the detailed binding mechanism of fentanyl. Surprisingly, in addition to the salt-bridge binding mode common to morphinan opiates, fentanyl can move deeper and form a stable hydrogen bond with the conserved His297[6.52], which has been suggested to modulate mOR's ligand affinity and pH dependence by previous mutagenesis experiments. Intriguingly, this secondary binding mode is only accessible when His297[6.52] adopts a neutral HID tautomer. Alternative binding modes may represent a general mechanism in G protein-coupled receptor-ligand recognition.

[1] Center for Drug Evaluation and Research, United State Food and Drug Administration, Silver Spring, MD, USA. [2] Department of Pharmaceutical Sciences, University of Maryland School of Pharmacy, Baltimore, MD, USA. ✉email: jana.shen@rx.umaryland.edu; christopher.ross.ellis@gmail.com

Opioids are highly effective pain relievers, but their addictive nature can easily lead to abuse and overdose-related deaths. From 1999 to 2018, almost 450,000 people died from opioid overdose in the United States[1]. Overdose deaths from synthetic opioids, represented by fentanyl and its derivatives, are now associated with more deaths than any other type of opioid[2]. The surge in fentanyl is attributed to high potency (50–400 times more potent than the naturally occurring morphine), fast onset, straightforward synthesis, and low-cost production[3–6]. Additionally, the fentanyl core is readily modified creating a vast chemical space of fentanyl analogs with abuse potential[7].

Fentanyl and morphine opioids produce strong analgesic responses through binding and subsequent activation of a class A G protein-coupled receptor (GPCR) μ-opioid receptor (mOR)[8]. In recent years, high-resolution crystal structures of mOR in complex with the morphinan agonist BU72[9], antagonist β-FNA[10], as well as the endogenous peptide analog agonist DAMGO[11] have been determined, featuring a salt bridge between a charged amine group of the ligand and a conserved residue Asp147 on the transmembrane helix (TM) 3 (Asp$^{3.32}$ in the Ballesteros-Weinstein numbering[12]) of mOR (Fig. 1a). The morphinan compounds and peptide analog also interact with a conserved His297 on TM6 (His$^{6.52}$ in the Ballesteros-Weinstein numbering[12]) via water-mediated hydrogen bonds. Mutagenesis studies demonstrated that mutation of either Asp147 or His297 as well as a reduced pH (which presumably protonates His297) decreases the binding affinities for DAMGO and naloxone (antagonist)[13–15].

Despite the importance, surprisingly little is known about the signaling mechanism of fentanyl and how it interacts with mOR to illicit analgesic response[6]. It is conceivable that fentanyl and its analogs bind and activate mOR in the same manner as morphinan compounds; however, the structural basis remains lacking. The aforementioned mutagenesis experiments performed to probe the role of Asp147 and His297 were inconclusive due to excessive non-specific binding of fentanyl[15] Docking[16,17]. and long-timescale molecular dynamics (MD) simulations[18] based on the docked structure of fentanyl in mOR confirmed the stability of the orthosteric binding mode involving the salt bridge with Asp147; however, the role of His297 has not been explored (Fig. 1).

Towards understanding the molecular mechanism of mOR activation by fentanyl, here we elucidate the detailed fentanyl–mOR binding mechanism by exploiting a morphinan-bound mOR crystal structure and several molecular dynamics (MD) methods, including the weighted ensemble (WE) approach[19–21] for enhanced path sampling and membrane-enabled continuous constant-pH MD (CpHMD) with replica-exchange[22–24]. The latter method has been previously applied to calculate p$K_a$s and describe proton-coupled conformational dynamics of membrane channels[25] and transporters[23,26,27]. Surprisingly, WE path sampling found that when His297 adopts the HID tautomer, fentanyl can move deeper into the mOR and establish an alternative binding mode through hydrogen bonding with His297. CpHMD titration showed that His297 favors the HIE tautomer in the apo mOR; however, interaction with the piperidine amine of fentanyl locks it in the HID tautomer. Additional microsecond equilibrium simulations were conducted to further verify the two binding modes and generate fentanyl–mOR interaction fingerprints. Alternative binding modes and involvement of tautomer states may represent general mechanisms in GPCR-ligand recognition. Our work provides a starting point for understanding how fentanyl activates mOR at a molecular level. Fentanyl analogs that can be significantly more

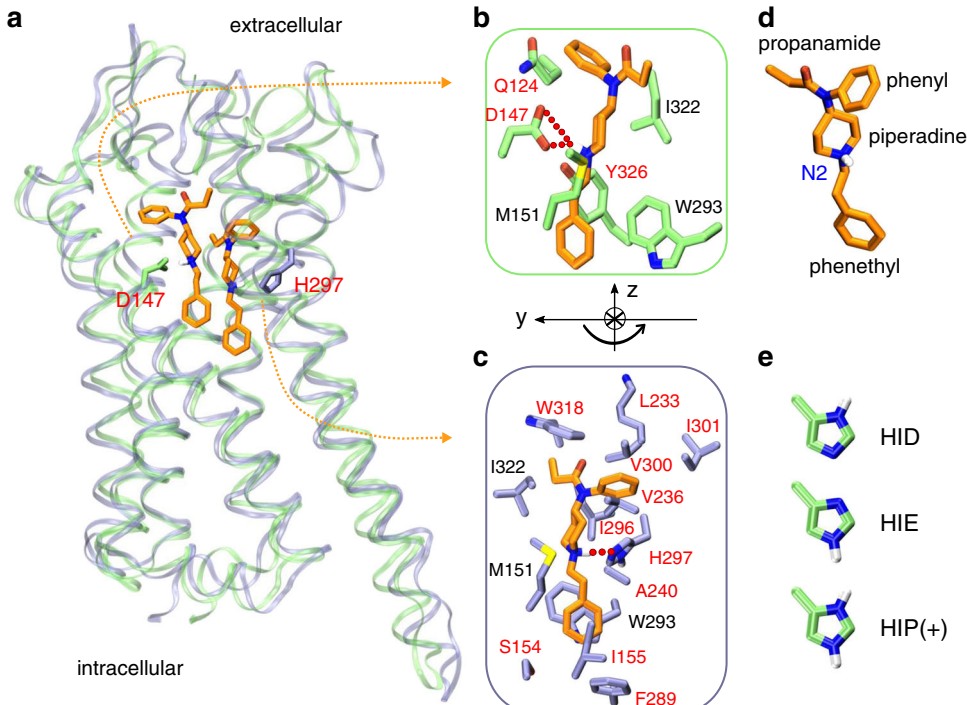

**Fig. 1 Fentanyl binding with mOR. a** Overlay of the representative simulation snapshots showing mOR is bound to fentanyl in the D147- (green) and H297- (purple) binding modes. **b** Zoomed-in view of the D147-binding mode, in which the charged piperidine amine of fentanyl forms a salt bridge with Asp147. **c** Zoomed-in view of the H297-binding mode, where the piperidine amine donates a proton to N$\epsilon$ of HID297. The curved arrow illustrates the change in the orientation of fentanyl in going from D147- to H297-binding mode. mOR residues making significant contacts with fentanyl (fraction greater than 0.5) are shown (see Fig. 4). Those unique to the two binding modes are labeled in red and otherwise in black. **d** Structure of fentanyl with different substituent groups and the protonated/charged amine labeled. **e** Histidine protonation states. HID and HIE are neutral while HIP is charged.

**Table 1 Summary of simulations in this work.**

| Simulation | Type | Starting configuration | | Time |
|---|---|---|---|---|
| | | Binding mode | H297 State | ($\mu$s) |
| WE-HIE | WE | D147 salt bridge | HIE | 24.3 |
| WE-HID | WE | D147 salt bridge | HID | 23.6 |
| CpH-apo | CpHMD | Apo active mOR | Dynamic | 0.32 |
| CpH-D147 | CpHMD | D147 salt bridge | Dynamic | 0.32 |
| CpH-H297 | CpHMD | H297 hydrogen bond | Dynamic | 0.32 |
| MD-D147(HID) | equil. MD | D147 salt-bridge | HID | 0.5 |
| MD-D147(HIE) | equil. MD | D147 salt-bridge | HIE | 0.5 |
| MD-D147(HIP) | equil. MD | D147 salt-bridge | HIP | 0.5 |
| MD-H297(HID) | equil. MD | H297 h-bond | HID | 1.0 |
| MD-H297(HIE) | equil. MD | H297 h-bond | HIE | 1.0 |
| MD-H297(HIP) | equil. MD | H297 h-bond | HIP | 1.0 |

potent and addictive are emerging on the dark market at a rapid pace. The molecular mechanism by which structural modifications alter fentanyl potency and abuse potential can inform the design of safer analgesics to combat the opioid crisis.

## Results

**Fentanyl unbinds from the D147-bound configurations in the presence of HIE297.** Following the 115-ns MD to relax the docked fentanyl–mOR complex (details see Methods and Protocols and Supplementary Fig. 1), we performed WE all-atom MD simulations to explore the detailed binding interactions of fentanyl in mOR (Table 1). The fentanyl RMSD was used as the progress coordinate. The MD trajectories were produced using the GPU-accelerated PMEMD engine in AMBER18[28] and the Python-based WESTPA tool[20] was used to control the WE protocol.

In the first WE simulation of 24 $\mu$s aggregate time, His297 was fixed in the HIE tautomer (N$\epsilon$ atom of imidazole is protonated), as in the recent mOR simulations by the Dror group[9,11]. The WE-HIE simulation proceeded as expected. In the first 75 iterations or 2.5 $\mu$s of cumulative sampling time, fentanyl's piperidine stays near Asp147, sampling both the salt-bridged and solvent-separated configurations, with the FEN–D147 distance (minimum heavy-atom distance between the piperidine amine and the carboxylate) below 5 Å (Fig. 2a and Supplementary Fig. 2). During this time, fentanyl $\Delta Z$ fluctuates between 7.5 and 10.5 Å, and RMSD stays below 4 Å (Supplementary Fig. 2). $\Delta Z$ is defined as the distance between the centers of mass of fentanyl and mOR in the z direction, whereby the N- (52–65) and C-terminal (336–347) residues of mOR were excluded from the calculation. After 75 iterations, fentanyl starts to move upward and away from Asp147; after about 140 iterations or 8 $\mu$s of cumulative sampling time, RMSD increases to above 7.5 Å and $\Delta Z$ starts to sample values above 14 Å, indicating that fentanyl is on the way to exit mOR (Supplementary Fig. 2). At the end of 300 iterations or 24 $\mu$s of cumulative sampling time, fentanyl reaches the extracellular end of mOR (Supplementary Fig. 2). It is noteworthy that in the WE-HIE simulation, the FEN–H297 distance from the piperidine nitrogen to the unprotonated imidazole nitrogen is always above 4 Å, indicating that fentanyl's piperidine does not form hydrogen bond interactions with His297 (Fig. 2a and Supplementary Fig. 2). Interestingly, even with the intact piperidine–D147 salt bridge, fentanyl can sample various configurations with a RMSD as high as 8 Å (Supplementary Fig. 2).

**Fentanyl samples both D147- and H297-bound configurations in the presence of HID297.** In addition to HIE, a neutral histidine can adopt the HID tautomer state, whereby the N$\delta$ atom is

protonated. Considering the important and yet unclear role of His297 in opioid-mOR binding, we conducted another WE simulation with His297 fixed in HID (WE-HID). Surprisingly, fentanyl did not exit mOR as was observed in the WE-HIE simulation. After about 27 iterations or 0.6 $\mu$s of cumulative sampling time, some of the trajectories start to sample configurations in which fentanyl laterally rotates 120°, translates 2 Å, and moves down 1 Å, enabling the formation of a stable hydrogen bond between the piperidine amine and the unprotonated N$\epsilon$ atom of HID297, (Figs. 1a, 2b, and Supplementary Fig. 3). At the same time, the RMSD remains below 7 Å. Unexpectedly, after about 210 iterations or 13 $\mu$s of cumulative sampling time, some trajectories start to sample configurations in which fentanyl is inserted deeper into the receptor (Supplementary Fig. 3). At the end of 20 $\mu$s aggregate time, fentanyl continues to sample the D147- and HID297-bound configurations along with positions in which it does not interact with either residue (Fig. 2b and Supplementary Fig. 3); however, the fentanyl $\Delta Z$ stay below 14 Å, indicating that it remains inside of the ligand accessible vestibule of mOR (Fig. 2b and Supplementary Fig. 3).

**Further comparison between the configurations from the WE-HIE and WE-HID simulations.** To further understand the differences in the configuration space sampled by fentanyl in the presence of HIE297 and HID297, we plotted FEN–H297 vs. FEN–D147 distance and color coded the data points by $\Delta Z$ of fentanyl. Corroborating with the previous analysis, these plots show that while the D147-bound configurations (FEN–D147 distance $\leq$ 3.5 Å) are sampled in both WE-HIE and WE-HID simulations, the H297-bound configurations (FEN–H297 distance $\leq$ 3.5 Å) are only sampled in the WE-HID simulation (Fig. 2c, d). Further, the H297-bound configurations sample lower $\Delta Z$ positions of 3–8 Å, as compared to the D147-bound configurations whereby $\Delta Z$ is in the range of 7–13 Å, (Fig. 2c–f, and Supplementary Fig. 2, 3). Interestingly, the WE-HID simulation also sampled fentanyl configurations deeply embedded in mOR ($\Delta Z$ $\leq$ 3 Å) but without a hydrogen bond with HID297 (FEN–H297 distance of 4–6 Å), suggesting that the piperidine–HID297 hydrogen bond may not be the only stabilizing factor for the deep insertion of fentanyl in mOR (Fig. 2d and Supplementary Fig. 4). Representative snapshots suggest that the interactions between the phenylethyl group and Trp293 may be a contributor (Fig. 2f).

**His297 favors the HIE tautomer in the apo mOR but the piperidine–HID297 interaction locks His297 in the HID state.** The WE simulations suggest that fentanyl has an alternative binding mode which may be promoted by the presence of the

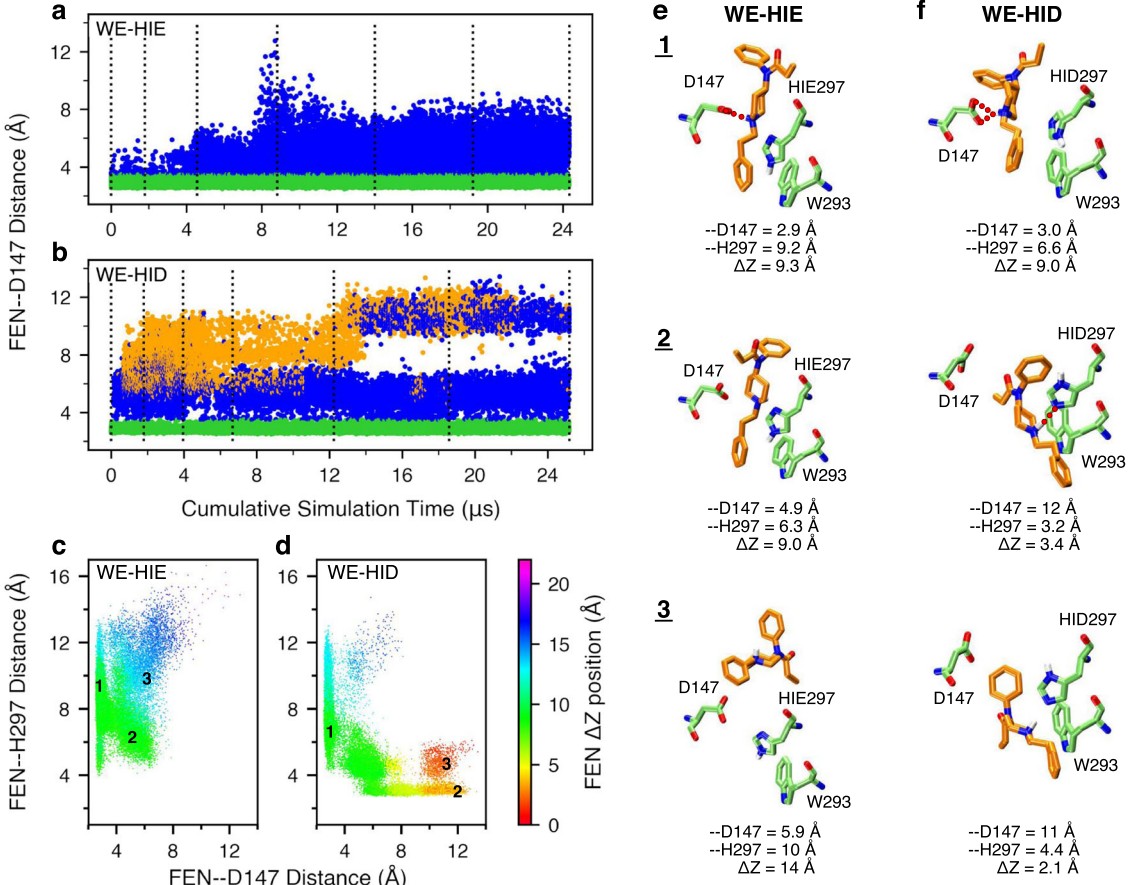

**Fig. 2 Fentanyl visits the D147-binding mode in the presence of HIE297 but both D147- and H297-binding modes in the presence of HID297. a, b** The FEN–D147 distance, referred to as the minimum distance between the piperidine nitrogen and the carboxylate oxygen of Asp147, as a function of the cumulative WE simulation time in the presence of HIE297 (**a**) or HID297 (**b**). Data with the FEN--D147 and FEN-H297 distances below 3.5 Å are colored green and orange, respectively, and otherwise blue. The unweighted data from all bins were taken and the time refers to the cumulative time. The dotted vertical lines are drawn at every 50 WE iterations. **c, d** FEN–H297 vs. FEN-D147 distance from the WE-HIE (**c**) and WE-HID (**d**) simulations. The data points are color coded by the fentanyl $\Delta Z$ position, defined as the distance between the centers of mass of fentanyl and mOR in the z direction, whereby the N- (52–65) and C-terminal (336–347) residues of mOR were excluded from the calculation. The FEN-H297 distance is measured between the piperidine nitrogen and the unprotonated imidazole nitrogen of His297. Three groups of data (labeled in the plots) taken from the last 50 iterations of each simulation were subjected to the hierarchical clustering analysis. For WE-HIE, the three groups were defined as FEN-D147 distance ≤ 3.5 Å; FEN-D147 distance ≥ 4 Å and FEN-H297 distance ≤ 8 Å; and FEN-D147 distance ≥ 4 Å and FEN-H297 distance ≥ 8 Å. For WE-HID, the three groups were defined as FEN-D147 distance ≤ 3.5 Å; FEN-H297 distance ≤ 3.5 Å; and FEN-D147 distance ≥ 8.5 Å and FEN-H297 distance ≥ 4 Å. **e, f** Representative structures of the most populated clusters from the WE-HIE (**e**) and WE-HID (**f**) data defined in (**c**) and (**d**). The FEN-D147 and –H297 distances and the fentanyl $\Delta Z$ position are given.

HID tautomer of His297. To determine the physiological relevance, we carried out titration simulations using the membrane-enabled hybrid-solvent CpHMD method with pH replica exchange[22,23] to determine the protonation state of His297 under physiological pH for the apo mOR and the fentanyl-bound mOR in the D147- as well as the H297-binding modes (Table 1). For each system, 16 pH replicas were simulated in the pH range 2.5–9.5, with the aggregate sampling time of 320 ns. All Asp/Glu/His and fentanyl's piperidine amine in the holo systems were allowed to titrate. The calculated p$K_a$ of His297 is well converged (Supplementary Fig. 4).

In the absence of ligand (CpH-apo simulation), the calculated macroscopic p$K_a$ of His297 is 6.8. At physiological pH 7.4, the HIE tautomer is predominantly sampled at 64%, while the HID tautomer and the charged HIP populations are 12% and 24%, respectively (Fig. 3a). The presence of fentanyl in the D147-binding mode upshifts the His297 p$K_a$ to 7.3 (CpH-D147 simulation). At physiological pH, both HIE and HIP are the predominant forms accounting for 39% and 44% of the

population, respectively, while HID accounts for 17% of the population (Fig. 3b).

Finally, CpHMD titration was also performed for the fentanyl–mOR complex in the H297-binding mode (CpH-H297 simulation). Interestingly, the calculated p$K_a$ of His297 is 6.7, nearly the same as for the apo mOR; however, at physiological pH HID is the predominant form with a population of 60%, while the HIE and HIP forms account for 20% each. Importantly, the protonation state of His297 is coupled to its distance to the piperidine amine of fentanyl (Fig. 3d). When the piperidine nitrogen is within 4 Å of the N$\epsilon$ atom of His297, the HID state is exclusively sampled, whereas the HIE and HIP states are only allowed when the piperidine–His297 distance is ≥7 Å (Fig. 3d). These data are consistent with the equilibrium MD which shows that the distance is 3.0 ± 0.22 Å, 7.4 ± 0.72 Å, and 8.0 ± 0.5 Å with HID297, HIE297, and HIP297, respectively, while the distance range 4–7 Å is rarely sampled (Supplementary Fig. 8). Note, in both holo simulations the piperidine amine remains protonated/charged in the entire pH range 2.5–9.5.

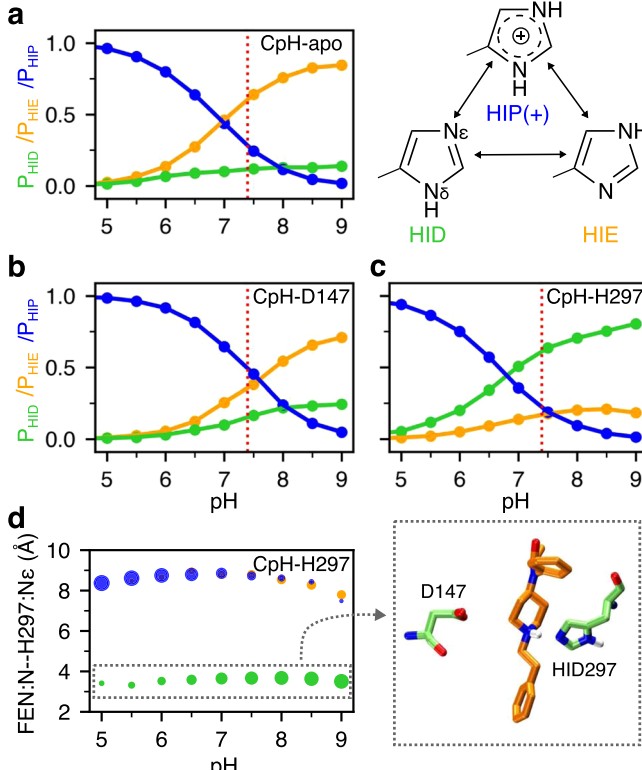

**Fig. 3 Protonation state of His297 is influenced by fentanyl binding.**
**a–c** Occupancies of the HID (green), HIE (orange), and HIP (blue) states of His297 as a function of pH from the replica-exchange CpHMD simulations of the apo mOR (**a**) and fentanyl-bound complex in the D147- (**b**) and H297-binding mode (**c**). The three states are in equilibrium through protonation/deprotonation and tautomerization. **d** Average distance between the piperidine nitrogen and His297's Nε at different pH conditions when His297 is in the HID (green), HIE (orange), or HIP (blue) state. The area of the data point is proportional to the occupancy of the protonation state. A zoomed-in snapshot corresponding to the HID state is shown.

The CpHMD simulations demonstrate that ligand interaction perturbs the protonation state of His297: while the apo mOR preferably samples HIE297, the population of HIP or HID state may increase upon ligand binding. When fentanyl interacts with Asp147, the HIP state is sampled with an equal probability as HIE, and when fentanyl interacts with His297, HID is the preferred state. These data provide an explanation as to why the H297-binding mode (quickly) emerged in the WE simulation with HID297 but not HIE297.

**Asp114 is deprotonated.** The protonation state of the highly conserved residue Asp114 (Asp[2.50]) in the active mOR remains unclear to this day. Despite not having a direct role in ligand binding, Asp114 is involved in mOR activation.[9,11,29,30] Previous experiments[30] and simulations[9,11,29] demonstrated that Asp114 binds a sodium ion in the inactive but not active state of GPCRs. Based on the lack of sodium binding, two previous MD studies used a protonated Asp114[9,11], while other published work did not specify the protonation state[18,31,32]. The CpHMD titration gave a $pK_a$ of $4.8 \pm 0.30$ for the apo and $5.1 \pm 0.26/0.29$ for the fentanyl-bound mOR in the D147- or H297-binding mode. Therefore, even though the $pK_a$s are upshifted relative to solution value of 3.8[33], Asp114 remains deprotonated at physiological pH in the active mOR according to the CpHMD simulations.

**The D147-binding mode is stable regardless of the protonation state of H297.** To further characterize fentanyl–mOR interactions and delineate the impact of the His297 protonation state, we carried out a series of equilibrium simulations (Table 1). First, three 0.5-$\mu$s simulations were initiated from the equilibrated fentanyl–mOR complex in the D147-binding mode with His297 fixed in the HID, HIE, and HIP states (Table 1). To quantify ligand-receptor interactions, the fractions of time for the mOR residues that form at least one heavy-atom contact with fentanyl were calculated (Fig. 4a–d, top panels). To determine what parts of fentanyl contribute to the receptor recognition, a fingerprint matrix was calculated which shows the contacts formed between specific mOR residues and fentanyl substituents (Fig. 4a–d, bottom panels). Simulations starting from the D147-binding mode demonstrated that many interactions are independent of the protonation state of His297. Most importantly, the piperidine–D147 salt bridge remains stable throughout the 0.5-$\mu$s trajectories with HIE297, HID297, and HIP297 (Supplementary Fig. 5 and Fig. 4a–c), consistent with the WE simulations. Interestingly, while maintaining the salt bridge with piperidine, Asp147 also interacts with phenyl and phenethyl at the same time (Fig. 4a–c, bottom panels), which may provide further stabilization to the D147-binding mode.

Another important fentanyl–mOR contact is the aromatic stacking interaction between the phenyl ring of the phenethyl group and Trp293 (Figs. 1b, 4a–c bottom panels, and Supplementary Fig. 6), which remains stable in all three simulations. The importance of the phenethyl group at this location in the 4-anilidopiperidine core of fentanyl is supported by the observations that substitution with methyl (as in N-methyl norfentanyl) increases the $K_i$ value by about 40 fold[34], and removal of one ethylene group renders the ligand inactive[35]. However, substitution with a different aromatic ring, e.g thiophene in sufentanil and ethyl tetrazolone in alfentanil, does not appear to be have a significant effect on binding affinity, although the latter ligands have an O-methyl group at the 4-axial hydrogen position[4]. The importance of the phenethyl-Trp293 stacking interaction is also consistent with a recent study which showed that removal of one methylene group from phenethyl increases the IC[50] value by two orders of magnitude[36].

**Fentanyl–mOR interaction profiles vary with different protonation state of His297 albeit in the same D147-binding mode.** Despite the similarities, the fentanyl–mOR interaction profiles obtained from the simulations MD-D147(HID), MD-D147 (HIE), MD-D147(HIP) show differences (Fig. 4a–c). To quantify the overall difference between two interaction profiles, the Tanimoto coefficient ($T_c$)[37] was calculated (Fig. 5a), where $T_c$ of 1 indicates that identical mOR residues are involved in binding to fentanyl. Accordingly, the contact profiles with HIE297 and HIP297 are more similar ($T_c$ of 0.81), whereas the contact profiles with HID297 and HIE297/HIP297 are somewhat less similar ($T_c$ of 0.71/0.73). As to the latter, the most significant differences are in the N-terminus. While fentanyl makes no contact with the N-terminus in the simulation with HID297, it interacts via propanamide and phenyl groups with His54 and Ser55 in the simulations with HIE297/HIP297. The fentanyl–N-terminus interactions are consistent with an experimental study which demonstrated that truncation of the mOR N-terminus increases the dissociation constant of fentanyl by 30 fold[38]. Significant differences are also seen in the TM2 contacts between simulations with HIE297 and HID297/HIP297. Four TM2 residues, Ala113, Asp114, Ala117, Gln124, are involved in stable interactions with fentanyl in the simulations with HID297/HIP297; however, only one TM2 residue Gln124 contacts fentanyl in the

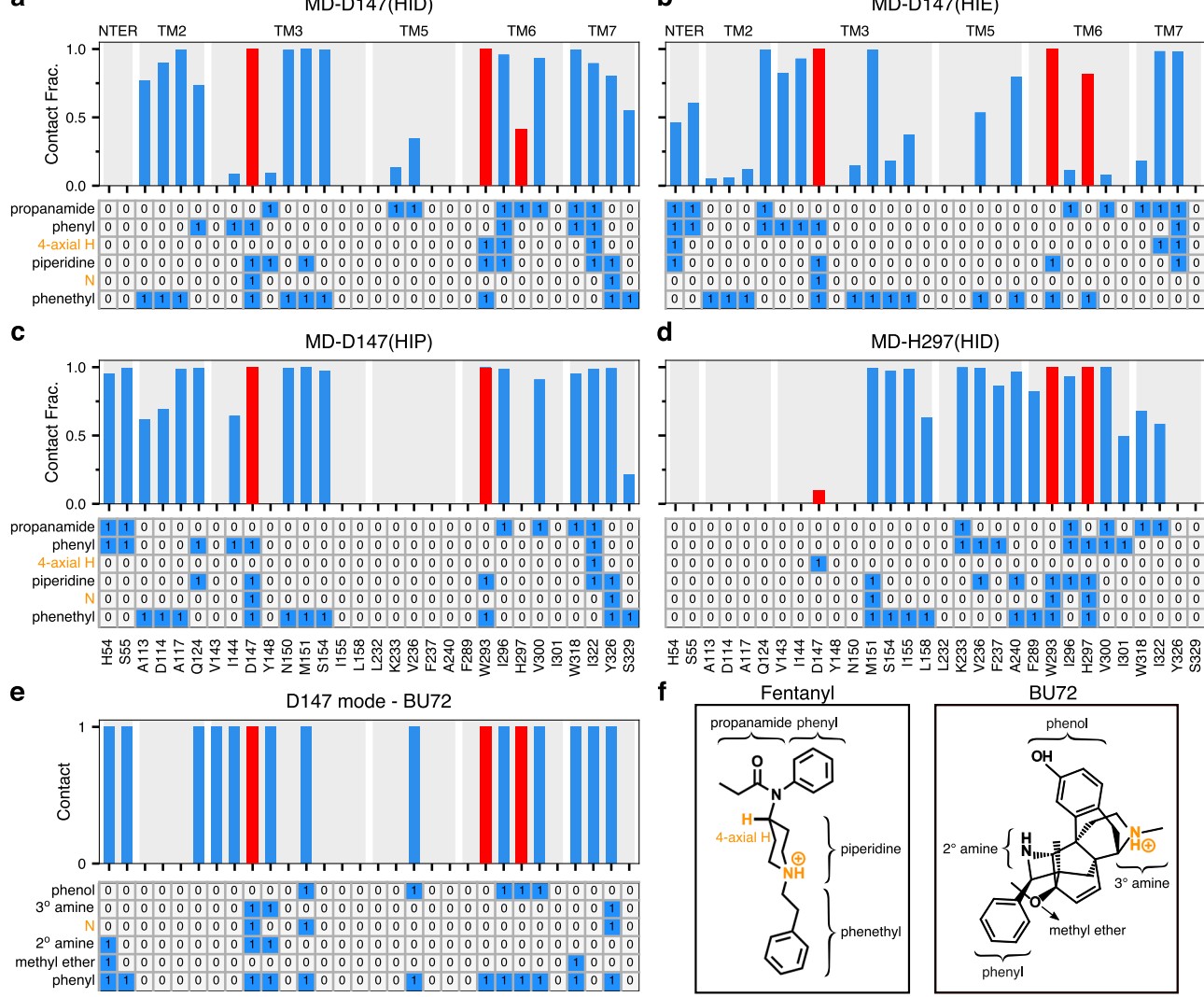

**Fig. 4 Fentanyl–mOR interaction profiles in the presence of different protonation state of His297 and comparison to the BU72-mOR contacts in the crystal structure. a–d** Top. Fraction of time that mOR residues form contacts with fentanyl in the equilibrium MD starting from the D147- (**a–c**) and H297-binding modes (**d**). A contact is considered formed if any sidechain heavy atom is within 4.5 Å of any fentanyl heavy atom. Only residues that form contacts for least 25% of the time in at least one of the six equilibrium simulations are shown. Contacts with Asp147, Trp293, and His297 are highlighted in red. **a–d** Bottom. Ligand-mOR fingerprint matrix showing the fentanyl groups as rows and mOR residue as columns. 1 represents in contact and 0 represents no contact. **e** mOR residues forming contacts with BU72 in the crystal structure (PDB: 5C1M[9]). **f** Chemical structure of fentanyl and BU72. Different substituent groups are labeled. The 4-axial hydrogen and amine nitrogen of the piperidine group are indicated.

simulation with HIE297 (Fig. 4a–c, top panels). A closer examination revealed that Gln124 interacts with phenyl in the simulation with HID297/HIP297 but it additionally interacts with propanamide in the simulation with HIE297, forming a stable hydrogen bond (Fig. 4a, b, bottom panels, Supplementary Fig. 7). This hydrogen bond may contribute to an upward shift of fentanyl's position in the simulation with HIE297 (see later discussion), resulting in a decrease of the aromatic stacking interaction between the phenyl ring of the phenethyl group and Trp293 (Supplementary Fig. 6).

**The H297-binding mode is stabilized by many fentanyl–mOR contacts in the presence of HID297.** To further evaluate the fentanyl–mOR interactions in the H297-binding mode, three 1-μs equilibrium simulations were initiated from the H297-binding mode with His297 fixed in the HID, HIE, and HIP states. We refer to these simulations as MD-H297(HID), MD-H297(HIE), and MD-H297(HIP), respectively (Table 1). In the MD-H297

(HID) simulation, the piperidine–H297 hydrogen bond remains stable; however, the hydrogen bond immediately breaks and the N–Nε distance fluctuates around 7.5 Å and 8.0 Å in the simulations with HIE297 and HIP297, respectively (Supplementary Fig. 8). These results are in agreement with the CpHMD titration, confirming that the H297-binding mode is only stable in the presence of HID297. In addition to the piperidine–H297 hydrogen bond, the simulation MD-H297(HID) shows that fentanyl forms stable contacts (with a contact fraction greater than 0.5) with over a dozen of residues on TM3, TM5, TM6, and TM7 (Figs. 1c, 4d), which explains the stability of the H297-binding mode in both MD-H297(HID) and WE-HID simulations.

**Comparison of the fentanyl–mOR contact profiles in the two binding modes.** Several interactions, e.g., stable contacts with Met151 (TM3), Trp293 (TM6), and Ile322 (TM7), are shared among all equilibrium simulations, regardless of the binding mode or His297 protonation state (Figs. 1b, c, 4a–d). Among

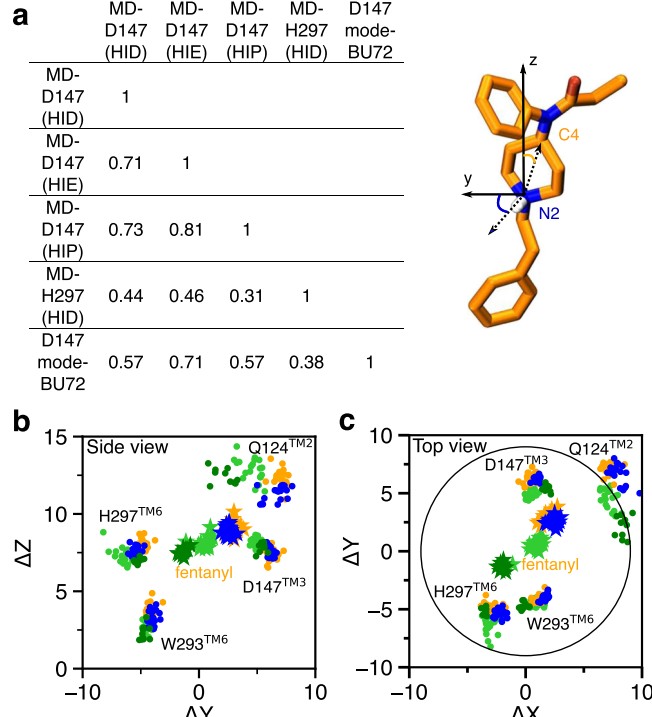

**Fig. 5 Contact similarity and spatial relationship between the D147- and H297-binding modes. a** Tanimoto coefficients ($T_c$) calculated using the binary contacts (details see Methods and Protocols). $T_c$ ranges from 0 to 1, where 1 indicates identical mOR residues are involved in fentanyl binding in both simulations. **b, c** Locations of fentanyl (stars) and critical amino acids (circles) plotted on the ($Y,Z$) and ($X,Y$) planes. Data points are sampled every 10 ns. The center of mass of mOR is set to origin. The data from MD-D147(HID), MD-D147(HIE), and MD-D147(HIP) are colored light green, orange, and blue, respectively, while the data from MD-H297(HID) are colored dark green. The $z$-axis is the membrane normal.

them is the aromatic stacking between the phenethyl group and Trp293 (Fig. 4a–d, Supplementary Fig. 6), which is stable in all simulations. Nonetheless, the contact profile from the simulation MD-H297(HID) is drastically different from those in the D147-binding mode. The $T_c$ value comparing MD-H297(HID) with MD-D147(HID), MD-D147(HIE), and MD-D147(HIP) are 0.44, 0.46, and 0.31, respectively (Fig. 5a). Fentanyl contacts with the N terminus and TM2 residues are completely absent and fewer TM7 residues are involved in fentanyl interactions in the simulation MD-H297(HID) (Fig. 4d). Most contacts uniquely observed for the H297-binding mode involve TM5 residues. As His297 is slightly below Asp147, a switch from the piperidine–D147 salt bridge to the piperidine–H297 hydrogen-bond results in a lower vertical position for fentanyl as compared to the D147-binding mode. The change in the vertical position likely contributes to the differences in the mOR residues interacting with fentanyl.

The fentanyl–H297 interactions also appear to perturb the local environment. While the backbone amide–carbonyl hydrogen bond between His297 and Trp293 is present in all equilibrium simulations, the backbone carbonyl of Trp293 also accepts a stable hydrogen bond from the Nδ atom of HID297 in the simulation MD-H297(HID) (Supplementary Fig. 7A, C). We hypothesize that the hydrogen-bond network (fentanyl–HID297–Trp293) together with the aromatic stacking between the phenethyl group and Trp293 contributes to a slight increase in the $\chi_2$ angle of Trp293 ($125 \pm 9°$), as compared to that in the D147-binding mode simulations ($112 \pm 10°$ with HIE297, $116 \pm 9.6°$ with HID297, and $110 \pm 9.5°$ with HIP297). Interestingly, the $\chi_2$ angle of Trp293 in

the X-ray structure of active mOR bound to BU72 (PDB: 5C1M[9]) is 120°, while that in the X-ray structure of the inactive mOR bound to the antagonist β-FNA (PDB: 4DKL[10]) is 80°.

Another intriguing feature of the simulation MD-H297(HID) is the transient contact between Asp147 and the 4-axial hydrogen of the piperidine ring (Fig. 4d, f). In the simulations of the D147-binding mode, the 4-axial hydrogen makes contact with TM6 or TM7 residues (Fig. 4a–c); however, in the simulation of the H297-binding mode, since the piperidine position is lower due to hydrogen bonding with His297, the 4-axial hydrogen position is also lowered, enabling an interaction with Asp147. Thus, we hypothesize that a substitution for a larger polar group at the 4-axial position might add stable interactions to both the D147- and H297-binding modes, which would potentially explain the increased binding affinity of fentanyl analogs with a methyl ester substitution at the 4-axial position, e.g. carfentanil and remifentanil[3,34,36,39].

**Comparison of the position and conformation of fentanyl in different binding modes.** The WE simulations demonstrated that the two fentanyl binding modes are readily accessible from one another in the presence of HID297 (Fig. 2d). The equilibrium simulations found that fentanyl contacts His297 in the D147-binding mode with HID297 or HIE297 (Fig. 4a, b) and it transiently interacts with Asp147 in the H297-binding mode with HID297 (Fig. 4d). To quantify the spatial relationship between the two binding modes, we calculated the center of mass (COM) positions of fentanyl and key contact residues relative to that of mOR based on the equilibrium simulations of the D147- and H297-binding modes and plotted in the ($Y,Z$) and ($X,Y$) planes. The resulting side (Fig. 5b) and top (Fig. 5c) views of the fentanyl and mOR residue locations showed that fentanyl adopts a similar position in the simulations of the D147-binding mode with HIE297 or HIP297; however, intriguingly, with HID297, the fentanyl position is moved towards the position it takes in the simulation of the H297-binding mode. Specifically, in going from the D147- to the H297-binding mode, fentanyl laterally rotates by about 120° such that the piperidine amine faces the Nε atom of His297, and translates by about 2 Å on the ($X,Y$) plane before moving down the $Z$-axis (Fig. 5b, c). The simulation MD-H297 (HID) gave the fentanyl $\Delta Z$ of $7.4 \pm 0.4$ Å, as compared to $8.3 \pm 0.4$, $8.8 \pm 0.3$, and $9.2 \pm 0.5$ from the simulations MD-D147(HID), MD-D147(HID), and MD-D147(HIP), respectively. Additionally, fentanyl is in a more upright position in the simulation of the H297 binding mode, with a vertical angle of 15°, compared to the angle of 20–40° in the simulations of the D147-binding mode (Supplementary Fig. 10).

A closer look at the conformation of His297 suggests that its $\chi_2$ dihedral angle may be modulated by the protonation/tautomer state (Supplementary Fig. 11). When His297 is in the HIE or HIP state, only the negative $\chi_2$ angle is sampled regardless of the binding mode; however, in the presence of HID297, the simulation of the D147-binding mode samples both negative and positive $\chi_2$ angles, whereas the simulation of the H297-binding mode only samples the positive $\chi_2$ angle. Thus, it is possible that HID297 allows both rotameric states, while the piperidine–H297 hydrogen bond locks the angle at 100°. These data further support the notion that the two binding modes are made accessible to one another in the presence of HID297.

**Comparison to the crystal structures of mOR in complex with BU72 and other ligands.** Finally, we compare the two fentanyl binding modes to the crystal structure of the BU72-bound mOR, which was used as a template to build the initial structure of fentanyl–mOR complex for WE simulations. The BU72-mOR

binding profile is most similar to fentanyl's D147-binding profile in the presence of HIE297 ($T_c$ of 0.71) due to their nearly identical interactions with the N-terminus, TM2, TM3 and TM7. To a lesser extent, the BU72 binding profile shares similarities with fentanyl's D147-binding profile in the presence of HID297 or HIP297 ($T_c$ of 0.57 for both). Importantly, the six residues, Gln124 (TM2), Asp147 and Met151 (TM3), Trp293 (TM6), Ile322 and Tyr326 (TM7), which form the foundation of the D147 binding pocket for fentanyl (with contact fraction greater than 0.5 regardless of the protonation state of His297) are present in the BU72-mOR binding contacts (Fig. 4a–c). In contrast, the BU72-mOR contact profile has a much lower overlap with fentanyl's H297-binding profile ($T_c$ of 0.38). In addition to BU72, we compare fentanyl's D147-binding profile (HIE297) to DAMGO- and β-FNA contacts with mOR based on the crystal structures (Supplementary Fig. 13). In contrast to fentanyl and BU72, DAMGO (a natural agonist) and β-FNA (an antagonist) do not form contacts with TM2 in the crystal structures. Other than that, DAMGO-mOR contact profile is similar to BU72-mOR and fentanyl's D147-binding profile (with HIE297), whereas β-FNA makes additional contacts with TM6 (Ala287, Ile290, and Val291) and has different contacts with TM7.

## Discussion

In summary, several molecular dynamics simulation techniques have been applied to investigate fentanyl binding to mOR. The WE simulations confirmed that fentanyl binds to mOR via the salt-bridge interaction between the piperidine amine and the conserved Asp147, consistent with the X-ray crystal structures of mOR in complex with BU72, β-FNA, and DAMGO[9–11]. However, surprisingly, when His297 is protonated at δ nitrogen (HID), fentanyl can also adopt a H297-binding mode, in which the piperidine amine donates a hydrogen bond to the ε nitrogen of HID297. The conventional single trajectory simulations confirmed that the D147-binding mode is stable regardless of the protonation state of His297, whereas the H297-binding mode is only compatible with HID297. These findings are consistent with a recent conventional MD study of the fentanyl–mOR binding, which found that the D147 binding mode was stable in the presence of HID297[18] but fentanyl moved deeper to contact HID297 in some trajectories (personal communication with Lipiński). Our CpHMD titration rationalized the finding by showing that in the absence of the piperidine amine-imidazole interaction, His297 can titrate via either Nδ or Nε; however, in the presence of the interaction, Nε loses the ability to gain a proton, locking histidine in the HID form.

We note that the calculation of the relative stability of the D147- vs. H297-binding mode is beyond the scope of the present work. Such a study would require converged WE simulations and accurate force field for quantifying the strengths of salt bridges and hydrogen bonds. Previous work by us[25] and others[40] showed that the CHARMM36[41] or the CHARMM36m force field[42] used in this work overstabilizes salt bridges formed by aspartates, although this might not be the case for the piperidine-Asp147 interaction. Overstabilization of salt bridges is a common problem with additive force fields, which may be overcome by explicit or implicit consideration of polarization[40]. We also note that given converged WE simulations, the transition rate between the two binding modes may be estimated[43], which is a topic of future exploration.

It is important to consider the physiological relevance of the H297-binding mode. According to the CpHMD titration, at physiological pH, HIE297 is the predominant form in the apo mOR, and HID297 is least populated in both the apo and holo mOR in the D147-binding mode. Consistent with these data, the crystal structure-based BU72-mOR contact profile bears the strongest resemblance to the simulated D147-binding mode with HIE297 as compared to HID297 or HIP297. Thus, we hypothesize that fentanyl primarily binds mOR via the D147 mode under physiological pH, while the H297-binding mode is a secondary state. This hypothesis is consistent with the recent experiments[15,44,45] showing that acidic pH has a negligible effect on fentanyl–mOR binding. These experiments also showed that fluorinated fentanyl which have lower $pK_a$s (6.8–7.2) than fentanyl (~8.9[46,47]) have increased affinities for mOR at lower pH. The CpHMD simulations showed that fentanyl's piperdine amine remains protonated/charged up to pH 9.5, while Asp147 is deprotonated with an estimated $pK_a$ of 3–4. Thus, our data are consistent with the hypothesis[44,45,48] that while fentanyl's D147-binding is not affected, lowering pH promotes protonation of the fluorinated fentanyls and thereby strengthening the salt bridge with Asp147. We expect the fluorinated fentanyls to have a lower potential for the His297-binding mode at physiological pH than fentanyl due to the decreased protonation of the piperidine.

The X-ray structures of mOR in complex with BU72, β-FNA, and DAMGO[9–11] show that while the piperidine amine forms a salt bridge with Asp147, the phenol hydroxyl group of the ligand forms a water-mediated hydrogen bond with His297. MD simulations of Dror and coworkers confirmed the stability of the water-mediated interactions between BU72 or DAMGO and His297 (HIE)[9,11]. Simulations of Carloni and coworkers[31] found that while in the D147-binding mode, the phenol group of morphine or hydromorphone forms a direct or water-mediated hydrogen bond with His297 (HIE), respectively. Morphine was also suggested to make hydrophobic contacts with His297 (HID) while in the D147-binding mode by the recent MD study of Lipiński and Sadlej[18]. The de novo binding simulations of the Filizola group[32] showed that oliceridine (TRV-130) which has an atypical chemical scaffold binds mOR via water-mediated interactions with Asp147, while frequently contacting His297 (protonation state unclear). Fentanyl does not have a phenol group, and it differs from morphinan ligands in several other ways. Fentanyl has an elongated shape; it is highly flexible with at least seven rotational bonds; and it has only two structural elements capable of forming hydrogen bonds (amine and carbonyl groups). In contrast, morphinan ligands are bulkier, rigid, and possess more structural elements (i.e. phenol group) with hydrogen bonding capabilities. The bulkier structure and additional hydrogen-bond interactions may further stabilize the piperidine-D147 salt bridge, preventing the ligand from moving deeper into mOR and access the H297-binding mode. Therefore, it is possible that the H297-binding mode is unique to fentanyl and analogs. Intriguingly, a combined MD and experimental study found that unlike synthetic antagonists, the endogenous agonist acetylcholine (a small elongated molecule) can diffuse into a much deeper binding pocket of M3 and M4 muscarinic acetylcholine receptors[49]. Thus, alternative binding modes may be a general phenomenon of GPCR-ligand recognition.

The CpHMD titration allowed us to determine the protonation states of His297 and all other titratable sites in mOR, including the conserved Asp114[2.50]. Sodium binding in the inactive mOR suggests a deprotonated Asp114[30], while the protonation state for the active mOR remains unclear. Recently, the $pK_a$ of the analogous Asp[2.50] in M2 muscarinic acetylcholine receptor (m2R) was calculated using the Poisson–Boltzmann method with a protein electric constant of 4[50]. The calculation gave a $pK_a$ of 8–12 when sodium is 5 Å away from Asp[2.50]. However, it is widely known that the continuum-based Poisson–Boltzmann methods overestimate the $pK_a$s of internal residues, particularly with a low dielectric constant (e.g., 4)[51]. The CpHMD simulations estimated the $pK_a$s of 4.8–5.1 for the apo and fentanyl-bound mOR, thus suggesting that it remains deprotonated in the active mOR.

Having a solution p$K_a$ of 6.5[33] and two neutral tautomer forms, histidine may sample all three protonation states in the protein environment at physiological pH 7.4. Our work demonstrates that the tautomer state of histidine in the ligand access region may alter the mechanism and possibly also the thermodynamics and kinetics of ligand binding. Thus, the conventional treatment in MD simulations, i.e., fixing histidine in a neutral tautomer state following the program default (HIE in Amber[28] and HID in CHARMM-GUI[52]) may not be appropriate for detailed investigations.

A caveat of the study is that all other histidines have been fixed in one protonation state in the WE and equilibrium simulations, even though some of them may sample alternative protonation state at physiological pH according to the CpHMD titration. A more complete understanding of how protonation states impact the conformational dynamics and ligand binding of GPCRs awaits the development of GPU-accelerated hybrid-solvent[22,23] and all-atom CpHMD methods[53] and their integration with enhanced sampling protocols such as the WE approach[19,21,54]. We also note that the present work is based on the activated structure of mOR and does not explore the large conformational changes of the receptor, which likely occur on a much slower timescale, e.g., the activation time of the class A GPCR $\alpha_{2A}$ adrenergic receptor was estimated as 40 ms[55]. Notwithstanding the caveats, our detailed fentanyl–mOR interaction fingerprint analysis provides a basis for pharmacological investigations of fentanyl analogs, particularly how structural modifications alter the binding properties of fentanyl derivatives which may have increased potency and abuse potential.

## Methods

All fixed-charge simulations (weighted ensemble and equilibrium molecular dynamics) were carried out using the GPU-accelerated *pmemd* engine in AMBER18[28]. The continuous constant pH molecular dynamics (CpHMD) simulations were carried out with the CHARMM program (version c42)[56]. The protein was represented by CHARMM36m[41] and CHARMM22/CMAP force fields[57,58] in the fixed-charge and CpHMD simulations, respectively. Water was represented by the CHARMM-style TIP3P force field.[56] The POPC and cholesterol molecules were represented by the CHARMM36 lipid force field[59]. The force field parameters for fentanyl were obtained using the ParamChem CGENFF server[60].

**Equilibration simulation of apo active mOR in a lipid bilayer**. The X-ray crystal structure of mOR in complex with BU72 (PDB ID: 5C1M)[9] was used as the starting model for apo active mOR. The crystal structure represents the wild type but contains a cysteine-s-acetamide (YCM) at position 57. This residue was converted to a cysteine (Cys57). A cholesterol molecule was resolved in the X-ray structure and is bound to the extracellular leaflet near TM7. This cholesterol and all crystal waters in the interior of mOR were kept. Seven additional water molecules were added using the DOWSER program[61]. Apo mOR was oriented with respect to membrane using the OPM (Orientations of Proteins in Membranes) database[62]. The CHARMM-GUI web server[52] was then used to construct the system of mOR embedded in a POPC (1-palmitoyl-2-oleoyl-glycero-3-phosphocholine) lipid bilayer. The disulfide bond observed in the crystal structure was imposed between Cys140 and Cys217. All titratable residues were fixed in the standard protonation states. All histidine residues were set to the neutral $N_\delta$ tautomer (HID) as in the default setting of CHARMM[56]. The system was first energy minimized using the steepest descent followed by conjugate gradient algorithm. The system was then gradually heated from 0 to 310 K over 200 ps with harmonic restraints on the protein heavy atoms, lipids, and bound water molecules (same as Step 1 in Supplementary Table 1). Following heating, the system was equilibrated for 117 ns, during which time the various harmonic restrains were gradually reduced to zero (Supplementary Table 1). The final dimension of the system was ~87 × 87 × 113 Å³. The final snapshot was used for constructing the fentanyl-bound mOR model and as the starting structure for the replica-exchange CpHMD simulations of apo mOR (CpH-apo).

**Relaxation of the docked fentanyl–mOR complex in a lipid bilayer**. The fentanyl-bound mOR model was prepared by superimposing a top fentanyl binding pose from a previous docking study[16] onto the final snapshot from the aforementioned equilibration simulation of apo active mOR. The docked pose showed a salt bridge between the piperidine nitrogen and Asp147 and an aromatic stacking between the phenethyl ring and His297. The docked model was equilibrated for 115 ns, using harmonic restraints imposed on the protein and fentanyl heavy atoms

in the first 5 ns (details see Supplementary Table 2). During the unrestrained part of the simulation, the root-mean-square deviation (RMSD) of the fentanyl heavy-atom positions with respect to the docked structure steadily increases in the first 20 ns and stabilizes at about 6 Å in the remainder of the 110 ns simulation (Supplementary Fig. 1). Concomitant with the fentanyl RMSD increase, the minimum distance between fentanyl's piperidine nitrogen and His297's imidazole nitrogen decreases from 10 to about 7 Å (Supplementary Fig. 1); however, the salt bridge between the positively charged piperidine amine and the negatively charged Asp147 remains largely stable except for occasional excursions (Supplementary Fig. 1), consistent with a previous $\mu$s simulation study[18]. The final snapshot was used as the starting configuration for the WE simulations.

**Weighted ensemble MD simulations**. Weighted ensemble (WE) is a path sampling protocol that uses splitting and merging trajectories to enhance sampling of rare events. Briefly, the configuration space is divided into bins based on a predetermined progress coordinate and a fixed number of walkers (trajectories) per bin is targeted. At the beginning of the simulation, walkers are initiated from a single bin and after a specified time interval, resampling is performed by evaluating the number of walkers per bin, and for bins with less than desired number of walkers, the walker is replicated (or split), and for bins with more than desired number of walkers, the walkers are pruned. Thus, over time, more bins are sampled and the simulation progresses along the progress coordinate. Details theory and algorithm can be found elsewhere[19,21,54].

Two WE MD simulations were carried out starting from the equilibrated fentanyl–mOR complex structure, with His297 fixed in either HIE (WE-HIE) or HID (WE-HID) state. The Python-based tool WESTPA[54] was used to control the WE protocol and data storage. The root-mean-square deviation (RMSD) of the fentanyl heavy-atom positions with respect to their starting positions was used as the progress coordinate. The configuration space was divided into bins that covered the RMSD values of 0 and 10 Å. A target number of four and five walkers per bin was used for the WE-HIE and WE-HID simulations, respectively. The fixed time interval for resampling each walker was 0.5 ns. The bin widths were changed manually in the beginning of the simulations to further accelerate sampling, and the final bin boundaries were placed at the following RMSD values: 0, 0.5, 1, 1.25, 1.5, 1.75, 2, 2.1, 2.2, 2.3, 2.4, 2.5, 2.6, 2.7, 2.8, 2.9, 3, 3.1, 3.2, 3.3, 3.4, 3.5, 3.6, 3.7, 3.8, 3.9, 4, 4.1, 4.2, 4.3, 4.4, 4.5, 4.6, 4.7, 4.8, 4.9, 5, 5.25, 5.5, 5.75, 6, 6.25, 6.5, 6.75, 7, 7.25, 7.5, 7.75, 8, 8.25, 8.5, 8.75, 9, 9.5, 10, >10. The WE simulations employed the Langevin thermostat, as a stochastic thermostat is required for the WE strategy to generate continuous pathways with no bias in the dynamics. A total of about 300 iterations were conducted for WE-HIE and WE-HID simulations. From the WE-HID simulation, we uncovered an alternative binding mode that involved a hydrogen bond between fentanyl's piperidine nitrogen and His297's N$\epsilon$. The bound pose was used as the starting configuration for the equilibrium simulation MD-H297(HID). To prepare the starting configurations for the equilibrium simulations MD-H297(HIE) and MD-H297(HIP), the protonation state of His297 in the bound pose was switched followed by 65 ns equilibration (Supplementary Table 3). The final snapshot which no longer contained the piperidine-H297(N$\epsilon$) hydrogen bond were used to start the MD-H297(HIE) and MD-H297(HIP) simulations.

**Continuous constant pH molecular dynamics (CpHMD)**. We applied the membrane-enabled hybrid-solvent continuous constant pH molecular dynamics (CpHMD) method[22,23] to determine the protonation states of all titratable residues in the mOR systems. In this method, conformational dynamics is propagated in explicit solvent and lipids, while the solvation forces for propagating titration coordinates are calculated using the membrane generalized-Born GBSW model[63] based on the conformations sampled in explicit solvent. To accelerate convergence of the coupled conformational and protonation-state sampling, a replica-exchange protocol in the pH space is used[22]. The membrane-enabled hybrid-solvent CpHMD method[22,23] has been validated for p$K_a$ calculations and pH-dependent simulations of transmembrane proteins[23,25,26]. The detailed protocols can be found here[24].

Three sets of replica-exchange CpHMD simulations were performed, starting from the equilibrated apo active mOR structure, equilibrated fentanyl–mOR complex in the D147-binding mode, and the fentanyl–mOR complex in the H297-binding mode obtained from the WE-HID simulation in which the FEN–H297 distance is less than 3.5 Å. The pH replica-exchange protocol included 16 replicas in the pH range 2.5–9.5 with an increment of 0.5 unit. A GB calculation was invoked every 10 MD steps to update the titration coordinates. In the GB calculation, the default settings were used, consistent with our previous work[22]. Each pH replica underwent molecular dynamics in the NPT ensemble with an aggregate sampling time of 320 ns. All Asp, Glu, and His sidechains as well as the piperidine amine were allowed to titrate. The model p$K_a$s of Asp, Glu, and His are 3.8, 4.2, and 6.5, respectively[33], while that of the piperidine amine in fentanyl is 8.9[46].

**Molecular dynamics protocol**. The temperature and pressure were maintained at 310 K and 1 atm by the Langevin thermostat and Monte Carlo barostat, respectively in the simulations with the Amber program[28], while the modified Hoover thermostat and Langevin piston coupling method were used in the simulations

with the CHARMM program[56]. Long-range electrostatics was treated by the particle-mesh Ewald (PME) method[64] with a real-space cut-off of 12 Å and a sixth-order interpolation with a 1.6-Å$^{-1}$ grid spacing. The van der Waals interactions were smoothly switched to zero between 10 and 12 Å. Bonds involving hydrogen atoms were constrained using the SHAKE algorithm to enable a 2-fs timestep. Analysis was performed using CPPTRAJ program[65]. For analysis of the equilibrium simulations starting from the D147- and H297-binding modes, the last 200 or 400 ns data were used, respectively.

**Clustering analysis**. The clustering analysis was performed using the cluster command in CPPTRAJ[65] with the hierarchical agglomerate algorithm. The distance between clusters was calculated based on the RMSD of fentanyl's heavy atoms. The distance cutoff was 3 Å.

**Calculation of Tanimoto coefficients**. Tanimoto coefficient between profiles A and B is calculated using the following equation:

$$S_{AB} = \frac{\sum_{i=1}^{n} x_{iA} \cdot x_{iB}}{\sum_{i=1}^{n} (x_{iA})^2 + \sum_{i=1}^{n} (x_{iB})^2 - \sum_{i=1}^{n} x_{iA} \cdot x_{iB}}, \quad (1)$$

where $x_{iA}$ or $x_{iB}$ denotes the contact value between fentanyl and residue $i$ of mOR from profile A or B, respectively. $x_i$ has a value of 1 if a contact exists and 0 otherwise.

**Binding site volume calculation**. A reviewer noted that BU72 is big and so the second binding mode may be the result of using the BU72-bound mOR crystal structure as a template to generate the initial structure for the fentanyl–mOR complex. The reviewer suggested running a 1-$\mu$s MD equilibration of the empty receptor (with the goal to "shrink the binding pocket"). To address this comment, we performed the binding site volume calculations using POVME2.0[66,67]. All structures were first aligned using the C$\alpha$ atoms of the binding site residues (Tyr75, Gln124, Asn127, Trp133, Ile144, Asp147, Tyr148, Met151, Phe152, Leu232, Lys233, Val236, Ala240, Trp293, Ile296, His297, Val300, Trp318, His319, Ile322, Tyr326) from the BU72-bound mOR crystal structure (PDB ID: 5C1M). All waters, ions, cholesterol, lipids, nanobody, and ligands were removed before the calculation. The binding pocket searching region is kept consistent throughout all systems by specifying an Inclusion box centered at the binding pocket center of mass (0,0,8) and sides of 12 Å, 12 Å, 15 Å in the $x$, $y$, and $z$ direction, respectively. For the volume calculation, trajectory snapshots were taken every 5 ns from the last 50 ns of the simulations.

The binding site volume based on the BU72-bound mOR crystal structure (with BU72 removed) is 371 Å$^3$. After 110 ns of MD equilibration before docking fentanyl, the volume is $359 \pm 12$ Å$^3$, which is similar to that of the crystal structure. To test if prolonged equilibration would shrink the binding site volume, we extended the simulation by 500 ns and found that the receptor's cavity volume increased to $465 \pm 51$ Å$^3$. An increase in volume can be rationalized as the result of relaxation and solvation of the binding site. Thus, a long MD equilibration of the empty receptor will not reduce the binding site volume, and the alternative binding mode of fentanyl is not a result of an expanded binding cavity due to the size of BU72.

**Reporting summary**. Further information on research design is available in the Nature Research Reporting Summary linked to this article.

## Data availability
Data supporting the findings of this manuscript are available from the corresponding authors upon reasonable request. A reporting summary for this article is available as a Supplementary Information file.

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

## Acknowledgements

The authors acknowledge financial support from FDA's Safety Research Interest Group and appointment to the Research Participation Programs at the Oak Ridge Institute for Science and Education through an interagency agreement between the Department of Energy and FDA. Financial support from the National Institutes of Health (GM098818) to J.S. and P.M. is also acknowledged.

## Author contributions

J.S. and C.R.E. designed the research. Q.N.V. performed the simulations with the help of P.M. Q.N.V., P.M., J.S., and C.R.E. analyzed the data. Q.N.V., J.S., and C.R.E. wrote the manuscript.

## Competing interests

The authors declare no competing interests.
