## [Peer Review File · Nature Communications]

Reviewer #1 (Remarks to the Author):

Quynh et.al used all-atom long time scaled MD simulations to elaborate the binding process of fentanyl against mu opioid-receptor. Fentanyl is a widely used painkiller and it has been widely studied before.

In my opinion, the biggest disadvantage of this paper is innovation. I found two similar works which were already published in 2019 as following:

(1) Molecular dynamics of fentanyl bound to μ -opioid receptor. J Mol Model. 2019 May 3;25(5):144. doi: 10.1007/s00894-019-3999-2

(2) Fentanyl Family at the Mu-Opioid Receptor: Uniform Assessment of Binding and Computational Analysis. Molecules. 2019 Feb; 24(4): 740.

The authors in above mentioned two papers also studied the binding mode of fentanyl against mu opioid receptor. They also used long time scaled MD simulations. The current work here shared a large common results with the above mentioned paper.

Another big concerns is for the pKa calculations which is still one of the biggest challenging in computational biology. I am not so sure how confident the authors about the exact pKa values obtained from MD simulation. It seems that the protonation state is also one of the major points in the work which will bring many concerns.

Some technical details:

(1) It is not clear to me how the author prepared the crystal structure mu-opioid receptor. In many cases, mutations will be introduced in the biological experiments to stabilize the crystals. Did the author mutate those residues back before MD simulations?

(2) Which lipids do the authors used for MD simulation? Was cholesterol added to the simulation system?

Reviewer #2 (Remarks to the Author):

In this study by Vo et al., the authors employ weighted ensemble (WE) and continuous constant pH (CpH) molecular dynamics (MD) simulations to analyse how fentanyl, a highly potent opioid, binds to the mu-opioid receptor (mOR). The authors identified two fentanyl binding modes, the latter of which has not been previously observed: (1) the D147 binding mode, in which the fentanyl piperidine amine forms a hydrogen bond (H-bond) with the D147 side chain, and (2) the H297 binding mode, only accessible when H297 is present as the HID tautomer, in which fentanyl protrudes further into mOR and its piperidine amine forms an H-bond with H297.

This study has important implications for our understanding of mOR-fentanyl/opioid interactions and highlights the importance of considering tautomerisation in ligand binding simulations. Overall, the manuscript is very interesting and well-presented. Additionally, the results and figures are well-

described. However, I feel the manuscript could be made stronger and more interesting for a wider audience by addressing the points below. In particular, the scope of the discussion could be expanded, and the significance of the results more clearly conveyed.

Major comments

1. A greater discussion of previous studies, which found that fentanyl binding and signalling are pH-independent (Spahn et al., *Science*, 2017; Spahn et al., *Scientific Reports*, 2018; Meyer et al., *Br J Pharmacol*, 2019) would be important (especially the former two studies, which do not report high non-specific binding of fentanyl).
2. The hierarchical clustering analysis (used for Fig 2C,D) is not included in the methods.
3. The authors discuss the sampling of the HID/HIE/HIP states when the fentanyl piperidine nitrogen atom is ≤ 4 Å and ≥ 7 Å from the His297 N ϵ (page 4) – what happens when the piperidine nitrogen is > 4 Å and < 7 Å away?
4. It feels slightly misleading when the authors refer to the contacts between fentanyl and residues in TM3,5-7 as conferring “the remarkable stability” to the H297-binding mode (page 5). Fentanyl also forms contacts with many TM3,5-7 residues in the D147-binding mode (as does BU72) (Fig. 4A-C,E). Perhaps this stance could be softened or clarified. The authors could also comment on the fact that most contacts uniquely observed for the fentanyl H297-binding mode appear to be with TM5 residues.
5. The manuscript includes a comparison of the fentanyl and BU72 binding contacts – why are other ligands with solved mOR-bound structures (beta-FNA, DAMGO) not considered?
6. A sizeable amount of the Concluding Discussion is devoted to Asp114, which is not/hardly mentioned in the Results. This section would read more fluidly if the CpHMD simulation results for Asp114 were included in the Results.
7. The claim “it is possible that the H297-binding mode is unique to fentanyl” (page 7) should be strengthened e.g. through more explanation/clarification of how the flexibility of fentanyl enables it to access the H297-binding mode (would the more rigid morphinan ligands be excluded if they are unable to flex in a certain manner?). Furthermore, how does the fentanyl H297-binding mode differ from the morphine binding mode, which forms an H-bond with H297, identified in (Cong et al., *PLOS ONE*, 2015)? Would this morphine binding mode also be classified as an H297-binding mode (thus detracting from the hypothesis that the H297-binding mode is unique to fentanyl)?
8. The manuscript would be strengthened by expanding the scope of the discussion (and if/where needed, analysis) to include points related to some of the following (for example):
 - When/how often is the H297-binding mode achieved? The authors point out that this mode is likely secondary to the D147-binding mode – is it likely to occur only in a negligible amount of cases? If so, would this explain the results of a previous study which found pH-independence for fentanyl signalling (Meyer et al., *Br J Pharmacol*, 2019)? Would there be any conditions (other than fentanyl binding) that favour HID tautomerisation and therefore the H297-binding mode?
 - What would the effects/consequences of the H297-binding mode be e.g. on receptor activation or

fentanyl potency?

- Expanded discussion of how modifications to fentanyl would (or would not) influence the H297-binding mode. Are there any modifications that would prevent the H297-binding mode? How would interpretations of previous results that the affinities of fentanyl derivatives, but not fentanyl, for mOR are pH-sensitive (Spahn et al., *Science*, 2017; Spahn et al., *Scientific Reports*, 2018) be influenced by the results presented here of the H297-binding mode?

Minor comments

1. It felt, to me, slightly contrived to introduce this study via a link to COVID-19. The opioid crisis is an enormous problem in its own right and, seeing as COVID-19 was not mentioned anywhere else in the text, I feel it could be removed from the Abstract/Introduction.

2. The ΔZ score should be more clearly explained when it is first introduced (in the “Fentanyl unbinds from...” paragraph; page 2, top right), e.g. using the explanation included in the Fig 2C,D legend.

3. The HIP tautomer should be defined more clearly in the text (it is mentioned suddenly at one point).

4. Amino acids are denoted in a mixture of single- and three-letter codes (e.g. His297 and H297); it could be preferable to remain consistent.

Figure comments

1. It could be helpful to move the structures of the histidine tautomers (Fig 3A) and fentanyl (Fig 4F) to the start of the manuscript (in Fig 1), as this information is highly relevant for understanding the all of the results.

2. For Fig S6, it could be helpful to add a figure showing the frequency of aromatic stacking (0-45° and 135-180° combined) vs. no aromatic stacking.

Alissa M Hummer

Reviewer #3 (Remarks to the Author):

In this manuscript, the authors investigated the binding interactions between fentanyl and μ -Opioid receptors using a combination of constant-pH and weighted ensemble simulations. They were able to identify the role of protonation of H297 and other conformational changes in the receptor responsible for fentanyl's changing binding affinity and binding sites. The paper is well written with sufficient details for reproduction of the results and was a pleasure to read. The results are a valuable demonstration of molecular insights that can be obtained for a complex biological process using constant-pH simulations, especially when coupled with an enhanced sampling technique such as the weighted ensemble strategy. Given the current importance of opioid receptors in public health and the compelling molecular insights provided about the binding interactions of these

receptors, I recommend publication of this manuscript in Nature Communications after minor revisions.

My suggestions for minor revisions to further improve the quality of the manuscript are the following:

1) Like many contemporary force fields, the CHARMM36 force field overstabilizes protein salt bridges (see Debiec et al. JCTC 2015), which may or may not significantly affect the binding of Fentanyl-D147. I would like to see a comment in the main text or supporting information on the potential effects of this overstabilization issue on the conclusions.

2) In the supplementary data (p. S-3), it is listed that “the simulation length for each walker was 0.5 ns”. To clarify, is the 0.5 ns the fixed time interval used for resampling in the weighted ensemble strategy? If so, I suggest referring to the 0.5 ns as the fixed time interval for resampling as “simulation length” may be confused with the length of the entire trajectory that consists of multiple fixed time intervals for resampling.

3) In the Methods, I assume that a Langevin thermostat was used since the simulations were carried out using the Amber software package. If this is the case, it is worth stating in the Methods that the simulations employed a stochastic thermostat and that this type of thermostat is required for the weighted ensemble strategy to generate continuous pathways with no bias in the dynamics.

4) Depending on the number of transition events simulated using WE MD, it may be possible to estimate rate constants for transitions between the two binding modes of HID297 using the approach from Suarez et al. JCTC (2014). This calculation is not essential for the manuscript, but would provide additional valuable insights about the relevant timescales if there is sufficient data to perform the calculations.

Reviewer #4 (Remarks to the Author):

The paper is very well written and nicely illustrated in the main text and also there is a lot of other informative data in the supplementary material. The employed methodologies (the weighted ensemble approach, continuous constant-pH MD with replica-exchange, and classical molecular dynamics) are valid and correctly used for establishing the binding modes and their stabilities of fentanyl in mOR.

Although the paper is interesting it presents only one compound, fentanyl, and no other more potent derivatives, although it is claimed in the paper that “Our work provides a basis for understanding mOR activation by diverse fentanyl derivatives”. The receptor activation is also not studied since the receptor structure is already activated.

The binding site for BU72 is very big since the compound is big so the presented double binding mode of fentanyl may be a result of increased binding site of mOR-BU72. It would be good to make a long MD equilibration (about 1 μ s) of empty receptor before docking of fentanyl.

We are pleased to submit a revised manuscript entitled “How μ -Opioid Receptor Recognizes Fentanyl” by Quynh N. Vo, Paween Mahinthichaichan, Jana Shen, and Christopher R. Ellis. We would like this work to be reconsidered for publication as a regular article in *Nature Communications*.

We thank the reviewers for their valuable critiques, comments as well as suggestions. We have made revision to the manuscript to fully address and accommodate them. Below we present our response and revision. The revised excerpt is in red and also highlighted in the revised manuscript. Unless otherwise noted, page, table, and figure numbers refer to the revised version.

Reviewer #1

Reviewer’s remark: *Quynh et.al used all-atom long time scaled MD simulations to elaborate the binding process of fentanyl against mu opioid-receptor. Fentanyl is a widely used painkiller and it has been widely studied before.*

Response: We thank the reviewer for recognizing the importance of fentanyl. The reviewer has correctly pointed out the use of all-atom long time scale MD; however, he/she has missed the two other novel computational methodologies used that have made our novel findings possible. Below we give our response and revision to the major and minor comments.

Major comments:

Comment 1: *In my opinion, the biggest disadvantage of this paper is innovation. I found two similar works which were already published in 2019 as following:*

(1) Molecular dynamics of fentanyl bound to mu-opioid receptor. J Mol Model. 2019 May 3;25(5):144. doi: 10.1007/s00894-019-3999-2

(2) Fentanyl Family at the Mu-Opioid Receptor: Uniform Assessment of Binding and Computational Analysis. Molecules. 2019 Feb; 24(4): 740.

The authors in above mentioned two papers also studied the binding mode of fentanyl against mu opioid receptor. They also used long time scaled MD simulations. The current work here shared a large common results with the above mentioned paper.

Response: We thank the reviewer for pointing out these two studies, one of which was cited in our original manuscript. We apologize for missing the second reference, which is now added in the revised manuscript in the context of discussing the fentanyl-mOR binding interactions.

Revision: Page 5, left column, first paragraph under Figure 3:

The importance of the phenethyl-Tyr293 stacking interaction is also consistent with a recent study which showed that removal of one methylene group from phenethyl increases the IC₅₀ value by two orders of magnitude (36).

Revision: Page 6, right column, last paragraph:

Ref. 36 is added.

Response: Regarding the lack of innovation, we respectfully disagree with the reviewer. First, we would like to point out the significant differences in the computational methodologies used in our work vs. the two referenced papers. In the cited papers by Lipiński, Sadlej et al., only conventional equilibrium molecular dynamics (MD) simulations were employed to investigate the binding of mOR with fentanyl and analogs. In the first paper (1), three replicates of 1.2- μ s conventional MD of fentanyl-mOR complex were conducted, and in the second paper (2), three replicates of 50-ns conventional MD were conducted for each mOR-fentanyl analog complex. All simulations assumed that the titratable sidechains are in the (CHARMM-GUI default) fixed protonation states.

By contrast, our work employed two state-of-the-art MD techniques. 1) Weighted Ensemble (WE) MD is a path sampling method for exploring rare events and larger conformational space. We conducted two sets of WE MD simulations (24 μ s each) in the presence of different tautomer states of H297. 2) The membrane-enabled continuous constant pH MD (CpHMD) is a method for studying proton-coupled conformational dynamics of transmembrane proteins. We conducted CpHMD to investigate the protonation state of H297 and how it impacts the fentanyl's binding mode.

Next, we would like to point out the new findings from our work which are not reported in the two referenced papers. While we confirmed the stability of the D147-binding mode as in the work of Lipiński, Sadlej et al., our WE MD simulations revealed an alternative binding mode of fentanyl in mOR, which has not been reported before. Furthermore, WE MD revealed that fentanyl's binding mode is dependent on the protonation state of the conserved H297, and CpHMD rationalized this dependence.

In sum, our work is novel, as it employed state-of-the-art MD based methodologies and offered new findings, none of which have been reported in the two publications by Lipiński, Sadlej et al. Finally, we would like to mention that Dr. Lipiński contacted us after seeing our manuscript in bioRxiv and noted that a similar H297 binding mode was also observed in their simulations but the discussion was removed from the publication under the pressure of the skeptical reviewers.

Above being said, to accommodate the reviewer's comment and in light of the communication with Dr. Lipiński, we made the following revision.

Revision: Page 8, left column, first paragraph:

These findings are consistent with a recent conventional MD study of the fentanyl-mOR binding, which found that the D147 binding mode was stable in the presence of HID297 (18) but fentanyl moved deeper to contact HID297 in some trajectories (personal communication with Lipiński).

Comment 2: *Another big concerns is for the pK_a calculations which is still one of the biggest challenging in computational biology. I am not so sure how confident the authors about the exact pK_a values obtained from MD simulation. It seems that the protonation state is also one of the major points in the work which will bring many concerns.*

Response: We agree with the reviewer that pK_a calculation is a challenging topic in computational biology. Traditional pK_a prediction methods are reliant on a single input structure. Over the past 15 years, we and others have been developing MD-based pK_a methods known as constant pH MD. Notably, the continuous constant pH MD (CpHMD) method which was first developed in the Brooks lab and later in the Shen lab demonstrated significantly improved accuracy for pK_a calculations over traditional methods (Alexov, Shen et al, Proteins 2011). Importantly, it has enabled for the first time simulations of proton-coupled conformational dynamics of proteins (Khandogin and Brooks, PNAS 2006; PNAS 2007). In more recent years, the method has been widely validated (Chen, Shen et al, Mol Simul 2014) and further developed to enable pK_a calculations and pH-dependent MD simulations of transmembrane proteins (Huang, Shen et al, Nat Commun 2016; Chen, Shen et al, J Phys Chem Lett; Henderson, Shen et al, PNAS 2020). The latter method, the membrane-

enabled CpHMD forms a basis of this work.

The reviewer is correct in that the protonation state is a major point in this work. It has been shown through both experiments and MD simulations that H297 plays an important role in mOR-ligand binding; however, the nature of the role remains unknown. Our work revealed a novel mechanism by which H297 can influence mOR-ligand binding and demonstrated the limitation of conventional MD simulations based on a single protonation state. The CpHMD method was briefly explained in the SI with proper references. To accommodate the reviewer's comment, we added a sentence about the membrane-enabled CpHMD method in Introduction.

Revision: Page 1, right column, last paragraph:

The latter method has been previously applied to calculate pK_a 's and describe proton-coupled conformational dynamics of membrane channels (25) and transporters (23,26,27).

Minor comments:

Comment 1: *It is not clear to me how the author prepared the crystal structure mu-opioid receptor. In many cases, mutations will be introduced in the biological experiments to stabilize the crystals. Did the author mutate those residues back before MD simulations?*

Response: The preparation of the simulation system using the crystal structure of μ OR (PDB 5C1M) is described in the SI:

"The crystal structure of mOR in complex with BU72 was used as the starting model for apo active mOR. All crystal waters in the interior of mOR were kept, and seven additional water molecules were added using the DOWSER program Apo mOR was oriented with respect to membrane using the OPM (Orientations of Proteins in Membranes) database..."

The crystal structure represents the wild type and contains a cysteine-s-acetamide (YCM) at position 57. This residue was converted to a cysteine (Cys57), and there was no further mutation. We added a description in the SI regarding this point.

Revision: SI page S-2, first paragraph:

The crystal structure represents the wild type mOR but contains a cysteine-s-acetamide (YCM) at position 57. This residue was converted back to a cysteine (Cys57).

Comment 2: *Which lipids do the authors used for MD simulation? Was cholesterol added to the simulation system?*

Response: POPC (1-palmitoyl-2-oleoyl-glycero-3-phosphocholine) lipid was used. There is one cholesterol molecule in the simulation system. This molecule was resolved in the crystal structure (PDB 5C1M) and is bound to the extracellular leaflet near TM7.

To clarify these points, we revised the corresponding sentences in the SI.

Revision: SI page S-2, first paragraph:

A cholesterol molecule was resolved in the X-ray structure and is bound to the extracellular leaflet near TM7. This cholesterol and all crystal waters in the interior of mOR were kept.

Revision: SI page S-2, first paragraph:

... mOR embedded in POPC (1-palmitoyl-2-oleoyl-glycero-3-phosphocholine) lipid bilayer.

Reviewer #2

Reviewer's remark: *In this study by Vo et al., the authors employ weighted ensemble (WE) and continuous constant pH (CpH) molecular dynamics (MD) simulations to analyse how fentanyl, a highly potent opioid, binds to the mu-opioid receptor (mOR). The authors identified two fentanyl*

binding modes, the latter of which has not been previously observed: (1) the D147 binding mode, in which the fentanyl piperidine amine forms a hydrogen bond (H-bond) with the D147 side chain, and (2) the H297 binding mode, only accessible when H297 is present as the HID tautomer, in which fentanyl protrudes further into mOR and its piperidine amine forms an H-bond with H297. This study has important implications for our understanding of mOR-fentanyl/opioid interactions and highlights the importance of considering tautomerisation in ligand binding simulations. Overall, the manuscript is very interesting and well-presented. Additionally, the results and figures are well-described. However, I feel the manuscript could be made stronger and more interesting for a wider audience by addressing the points below. In particular, the scope of the discussion could be expanded, and the significance of the results more clearly conveyed.

Response: We thank the reviewer for the favorable view of the paper and the detailed comments. Below we give our response and revision.

Major comments:

Comment 1: A greater discussion of previous studies, which found that fentanyl binding and signalling are pH-independent (Spahn et al., *Science*, 2017; Spahn et al., *Scientific Reports*, 2018; Meyer et al., *Br J Pharmacol*, 2019) would be important (especially the former two studies, which do not report high non-specific binding of fentanyl).

Response: To address this comment, we expanded the discussion.

Revision: Page 8, left column, third paragraph:

This hypothesis is consistent with the recent experiments (15,44,45) showing that acidic pH has a negligible effect on fentanyl-mOR binding. These experiments also showed that fluorinated fentanyl which have lower pK_a 's (6.8–7.2) than fentanyl (~8.9) (46,47) have increased affinities for mOR at lower pH. The CpHMD simulations showed that fentanyl's piperidine amine remains protonated/charged up to pH 9.5, while Asp147 is deprotonated with an estimated pK_a of 3–4. Thus, our data is consistent with the hypothesis (44,45,48) that while fentanyl's D147-binding is not affected, lowering pH promotes protonation of the fluorinated fentanyls and thereby strengthening the salt bridge with Asp147. We expect the fluorinated fentanyls to have a lower potential for the His297-binding mode at physiological pH than fentanyl due to the decreased protonation of the piperidine.

Comment 2: The hierarchical clustering analysis (used for Fig 2C,D) is not included in the methods.

Response: Description of the clustering method is now added to the SI.

Revision: SI page S-5, bottom:

The clustering analysis was performed using the cluster command in CPPTRAJ program with the hierarchical agglomerate algorithm. The distance between clusters was calculated based on the RMSD of fentanyl's heavy atoms. The distance cutoff was 3 Å.

Comment 3: The authors discuss the sampling of the HID/HIE/HIP states when the fentanyl piperidine nitrogen atom is ≤ 4 Å and ≥ 7 Å from the His297 $N\epsilon$ (page 4) – what happens when the piperidine nitrogen is > 4 Å and < 7 Å away?

Response: As can be seen from Figure 3D, fentanyl rarely samples distances of 4–7 Å between its piperidine amine and His297 residue. This observation is consistent with equilibrium MD simulations in which the distance is 3.0 ± 0.22 Å when His297 is in the HID tautomer, or 7.4 ± 0.72 Å and 8.0 ± 0.5 Å when His297 is in HIE or HIP form, respectively (Figure S8). We added the clarification in the main text.

Revision: Page 3, left column, second paragraph:

The CpHMD data is consistent with the equilibrium MD which shows that the distance is 3.0 ± 0.22 Å, 7.4 ± 0.72 Å, and 8.0 ± 0.5 Å with HID297, HIE297, and HIP297, respectively, while the distance range 4–7 Å is rarely sampled (Fig. S8). Note, in both holo simulations the piperidine amine remains protonated/charged in the entire pH range 2.5–9.5.

Comment 4: *It feels slightly misleading when the authors refer to the contacts between fentanyl and residues in TM3,5-7 as conferring “the remarkable stability” to the H297-binding mode (page 5). Fentanyl also forms contacts with many TM3,5-7 residues in the D147-binding mode (as does BU72) (Fig. 4A-C,E). Perhaps this stance could be softened or clarified. The authors could also comment on the fact that most contacts uniquely observed for the fentanyl H297-binding mode appear to be with TM5 residues.*

Response: We deemed the H297-binding mode as “remarkably stable” because all the contacts within H297-binding mode, except with Asp147, have a contact fraction greater than 0.5. Nonetheless, we have softened the language used in the text to describe the H297-binding mode and commented on TM5 contacts as suggested by the reviewer.

Revision: Page 5, right column, second paragraph:

... the simulation MD-H297(HID) shows that fentanyl forms stable contacts (with a contact fraction greater than 0.5) with over a dozen of residues ..., which explains the stability of the H297-binding mode ...

Revision: Page 5, right column, last paragraph:

Most contacts uniquely observed for the H297-binding mode involve TM5 residues.

Comment 5: *The manuscript includes a comparison of the fentanyl and BU72 binding contacts – why are other ligands with solved mOR-bound structures (beta-FNA, DAMGO) not considered?*

Response: We only compared the binding profile of fentanyl with the BU72-mOR complex structure, as it was used as a template to build the initial structure of fentanyl-mOR complex for WE simulations. We also note that both β -FNA (Huang et al., Nature 2015) and DAMGO (Koehl et al., Nature 2018) have been shown to occupy the same binding pocket and adopt similar bound poses as BU72.

To clarify these points, we made revision to the main text and added a new contact plot in the SI that compares fentanyl-mOR binding profile with the contact profiles of BU72, DAMGO, and β -FNA based on the X-ray structures.

Revision: Page 7, right column, third paragraph:

Comparison to the X-ray structures of mOR in complex with BU72 and other ligands. Finally, we compare the two fentanyl binding modes to the crystal structure of the BU72-bound mOR, which was used as a template to build the initial structure of fentanyl-mOR complex for WE simulations...

In addition to BU72, we compare fentanyl's D147-binding profile (HIE297) to DAMGO- and β -FNA contacts with mOR based on the crystal structures (Fig. S13). In contrast to fentanyl and BU72, DAMGO (a natural agonist) and β -FNA (an antagonist) do not form contacts with TM2 in the crystal structures. Other than that, DAMGO-mOR contact profile is similar to BU72-mOR and fentanyl's D147-binding profile (with HIE297), whereas β -FNA makes additional contacts with TM6 (Ala287, I290, and V291) and has different contacts with TM7.

Revision: SI Page S-21:

Figure S13 is added.

Comment 6: *A sizeable amount of the Concluding Discussion is devoted to Asp114, which is not/hardly mentioned in the Results. This section would read more fluidly if the CpHMD simulation*

results for Asp114 were included in the Results.

Response: We added a discussion regarding Asp114 in Results and Discussion and significantly shortened the related discussion in Concluding Discussion.

Revision: Page 3, right column, last paragraph:

Asp114 is deprotonated. The protonation state of the highly conserved residue Asp114 (Asp²⁻⁵⁰) in the active mOR remains unclear to this day. Despite not having a direct role in ligand binding, Asp114 is involved in mOR activation (9,11,29,30). Previous experiments (30) and simulations (9,11,29) demonstrated that Asp114 binds a sodium ion in the inactive but not active state of GPCRs. Based on the lack of sodium binding, two previous MD studies used a protonated Asp114 (9,11), while other published work did not specify the protonation state (18,31,32). The CpHMD titration gave a pK_a of 4.8 ± 0.30 for the apo and $5.1 \pm 0.26/0.29$ for the holo active mOR in the D147- or H297-binding mode. Therefore, even though the pK_a 's are upshifted relative to the solution value of 3.8 (33), Asp114 remains deprotonated at physiological pH in the active mOR according to the CpHMD simulations.

Revision: Page 8, right column, second paragraph:

The CpHMD titration allowed us to determine the protonation states of His297 and all other titratable sites in mOR, including the conserved Asp114²⁻⁵⁰. Sodium binding in the inactive mOR suggests a deprotonated Asp114 (30), while the protonation state for the active mOR remains unclear...

The CpHMD simulations estimated the pK_a 's of 4.8–5.1 for the apo and fentanyl-bound mOR, thus suggesting that it remains deprotonated in the active mOR.

Comment 7: *The claim “it is possible that the H297-binding mode is unique to fentanyl” (page 7) should be strengthened e.g. through more explanation/clarification of how the flexibility of fentanyl enables it to access the H297-binding mode (would the more rigid morphinan ligands be excluded if they are unable to flex in a certain manner?). Furthermore, how does the fentanyl H297-binding mode differ from the morphine binding mode, which forms an H-bond with H297, identified in (Cong et al., PLOS ONE, 2015)? Would this morphine binding mode also be classified as an H297-binding mode (thus detracting from the hypothesis that the H297-binding mode is unique to fentanyl)?*

Response: Our work showed that fentanyl can bind via either D147 or H297 but not at the same time. This is in contrast to the crystal structures and simulations of morphinan ligands, which bind mOR via the D147-binding mode, while forming a water-mediated hydrogen bond with H297 at the same time (Cong et al, PLOS ONE, 2015; Huang et al, Nature 2915; Koehl et al, Nature 2018).

To clarify this point and expand on the discussion of the differences between fentanyl and morphinan ligands which may lead to the unique H297 binding mode, we made the following revision.

Revision: Page 8, left column, last paragraph:

The X-ray structures of mOR in complex with BU72, β -FNA, and DAMGO (9-11) show that while the piperidine amine forms a salt bridge with Asp147, the phenol hydroxyl group of the ligand forms a water-mediated hydrogen bond with His297. MD simulations of Dror and coworkers confirmed the stability of the water-mediated interactions between BU72 or DAMGO and His297 (HIE). (9,11) Simulations of Carloni and coworkers (31) found that while in the D147-binding mode, the phenol group of morphine or hydromorphone forms a direct or water-mediated hydrogen bond with His297 (HIE), respectively. Morphine was also suggested to make hydrophobic contacts with His297 (HID) while in the D147-binding mode by the recent MD study of Lipiński and Sadlej. (18) The de novo binding simulations of the Filizola group (32) showed that oliceridine (TRV-130) which has an atypical chemical scaffold binds mOR via water-mediated interactions with Asp147, while frequently contacting His297 (protonation state unclear). Fentanyl does not have a phenol group, and it differs from morphinan ligands in several other ways. Fentanyl has an elongated shape;

it is highly flexible with at least seven rotational bonds; and it has only two structural elements capable of forming hydrogen bonds (amine and carbonyl groups). In contrast, morphinan ligands are bulkier, rigid, and possess more structural elements (i.e. phenol group) with hydrogen bonding capabilities. The bulkier structure and additional hydrogen bond interactions may further stabilize the piperidine-D147 salt bridge, preventing the ligand from moving deeper into mOR and access the H297-binding mode. Therefore, it is possible that the H297-binding mode is unique to fentanyl and analogs.

Comment 8: *The manuscript would be strengthened by expanding the scope of the discussion (and if/where needed, analysis) to include points related to some of the following (for example):*

- *When/how often is the H297-binding mode achieved? The authors point out that this mode is likely secondary to the D147-binding mode – is it likely to occur only in a negligible amount of cases? If so, would this explain the results of a previous study which found pH-independence for fentanyl signalling (Meyer et al., Br J Pharmacol, 2019)? Would there be any conditions (other than fentanyl binding) that favour HID tautomerisation and therefore the H297-binding mode?*
- *What would the effects/consequences of the H297-binding mode be e.g. on receptor activation or fentanyl potency?*
- *Expanded discussion of how modifications to fentanyl would (or would not) influence the H297-binding mode. Are there any modifications that would prevent the H297-binding mode? How would interpretations of previous results that the affinities of fentanyl derivatives, but not fentanyl, for mOR are pH-sensitive (Spahn et al., Science, 2017; Spahn et al., Scientific Reports, 2018) be influenced by the results presented here of the H297-binding mode?*

Response to comment 8.1: Our CpHMD simulations showed that for the apo mOR, HIE is the dominant state at physiological pH, whereas the population for HID is only 12%. That's the reason why we hypothesized that the H297-binding mode is secondary to the D147-binding mode. However, the WE simulation was not long enough in order for us to estimate the relative free energy of the two binding modes. To clarify this point, we added a discussion.

The reviewer is correct in that the D147-binding mode being primary is consistent with the experiment of Meyer, Stein, et al (Meyer et al., Br J Pharmacol, 2019), which showed that fentanyl has a similar affinity for mOR at pH 6, 6.5 and 7.4, although the fraction of unspecific binding is very high. As to whether there is a condition that would favor HID tautomer and thereby the H297-binding mode, we do not know the answer and we do not think it is appropriate to make a speculation in the absence of any data.

Revision: Page 8, left column, second paragraph:

We note that calculation of the relative stability of the D147- vs. H297-binding mode is beyond the scope of the present work. Such a study would require converged WE simulations and accurate force field for quantifying the strengths of salt bridges and hydrogen bonds...

Response to comment 8.2: We did not observe any significant changes to the receptor conformation during the 1- μ s equilibrium MD simulation of the H297-binding mode. That said, if there was any effect, it would likely occur on a much longer timescale. Thus, we cannot comment on the effect of the H297-binding mode on receptor activation. We added a brief discussion to reflect this point. We would need free energy data in order to comment on the potency, but it is beyond the scope of this work.

Revision: Page 9, left column, first paragraph:

We also note that the present work is based on the activated structure of mOR and does not explore the large conformational changes of the receptor, which likely occur on a much slower timescale, e.g., the activation time of the class A GPCR α_{2A} adrenergic receptor was estimated as 40 ms (55).

Response to comment 8.3: Stein and coworkers hypothesized that the pH-sensitive affinities

of the fentanyl derivatives FF3 and NFEPP are due to the pH-modulated piperidine-D147 salt bridge interaction. Specifically, these compounds (piperidine amines) have solution pK_a 's below physiological pH 7.4, and thus the salt bridge is intact at low pH but may be weakened or disrupted at physiological pH. Deprotonation of piperidine would prevent it from donating a hydrogen bond to H297. Thus, we expect these compounds to have a lower potential to access the H297-binding mode at physiological pH.

Revision: Page 8, left column, second paragraph from the bottom:

This hypothesis is consistent with the recent experiments (15,44,45) showing that acidic pH has a negligible effect on fentanyl-mOR binding. These experiments also showed that fluorinated fentanyl which have lower pK_a 's (6.8–7.2) than fentanyl (~8.9) (46,47) have increased affinities for mOR at lower pH. The CpHMD simulations showed that fentanyl's piperidine amine remains protonated/charged up to pH 9.5, while Asp147 is deprotonated with an estimated pK_a of 3–4. Thus, our data is consistent with the hypothesis (44,45,48) that while fentanyl's D147-binding is not affected, lowering pH promotes protonation of the fluorinated fentanyls and thereby strengthening the salt bridge with Asp147. We expect the fluorinated fentanyls to have a lower potential for the His297-binding mode at physiological pH than fentanyl due to the decreased protonation of the piperidine.

Minor comments:

Comment 1: *It felt, to me, slightly contrived to introduce this study via a link to COVID-19. The opioid crisis is an enormous problem in its own right and, seeing as COVID-19 was not mentioned anywhere else in the text, I feel it could be removed from the Abstract/Introduction.*

Response: To accommodate the reviewer's comment, we have removed the text related to COVID-19 from the Abstract and Introduction.

Revision: Page 1, left column, Abstract:

Drug overdose has claimed over 70,000 lives in the United States in 2019.

Revision: Page 1, left column, first paragraph:

From 1999–2018, almost 450,000 people died from opioid overdose in the United States (1).

Comment 2: *The ΔZ score should be more clearly explained when it is first introduced...*

Response: We revised the definition as suggested by the reviewer.

Revision: Page 2, left column, second paragraph:

ΔZ is defined as the distance between the centers of mass (COM) of fentanyl and mOR in the z direction, whereby the N- (52–65) and C-terminal (336–347) residues were excluded from the calculation.

Comment 3: *The HIP tautomer should be defined more clearly in the text (it is mentioned suddenly at one point).*

Response and Revision: HIP tautomer is now better defined in the updated Figure 1.

Comment 4: *Amino acids are denoted in a mixture of single- and three-letter codes (e.g. His297 and H297); it could be preferable to remain consistent.*

Response and Revision: The text has been edited throughout to be consistent.

Figure comments:

Comment 1: *It could be helpful to move the structures of the histidine tautomers (Fig 3A) and fentanyl (Fig 4F) to the start of the manuscript (in Fig 1), as this information is highly relevant for understanding the all of the results.*

Response and Revision: Figure 1 has been updated as suggested by the reviewer.

Comment 2: *For Fig S6, it could be helpful to add a figure showing the frequency of aromatic stacking (0-45° and 135-180° combined) vs. no aromatic stacking.*

Response and Revision: Figure S6 has been revised to include a new panel as suggested by the reviewer.

Reviewer #3

Reviewer's remark: *In this manuscript, the authors investigated the binding interactions between fentanyl and mu-Opioid receptors using a combination of constant-pH and weighted ensemble simulations. They were able to identify the role of protonation of H297 and other conformational changes in the receptor responsible for fentanyl's changing binding affinity and binding sites. The paper is well written with sufficient details for reproduction of the results and was a pleasure to read. The results are a valuable demonstration of molecular insights that can be obtained for a complex biological process using constant-pH simulations, especially when coupled with an enhanced sampling technique such as the weighted ensemble strategy. Given the current importance of opioid receptors in public health and the compelling molecular insights provided about the binding interactions of these receptors, I recommend publication of this manuscript in Nature Communications after minor revisions.*

Response: We thank the reviewer for the favorable view of the paper and the helpful comments/suggestions. Below we give our response and revision to the minor comments.

Minor comments:

Comment 1: *Like many contemporary force fields, the CHARMM36 force field overstabilizes protein salt bridges (see Debiec et al. JCTC 2015), which may or may not significantly affect the binding of Fentanyl-D147. I would like to see a comment in the main text or supporting information on the potential effects of this overstabilization issue on the conclusions.*

Response: We agree with the reviewer that the CHARMM36 and CHARMM36m (used in the fixed-charge simulations of the present work) force field overstabilize salt bridges. We added a brief discussion with a citation to the paper by Debiec, Chong, et al.

Revision: Page 8, left column, second paragraph:

Previous work by us (25) and others (40) showed that the CHARMM36 (41) or the CHARMM36m force field (42) used in this work overstabilizes salt bridges formed by aspartates, although this might not be the case for the piperidine-Asp147 interaction. Overstabilization of salt bridges is a common problem of additive force fields, which may be overcome by explicit or implicit consideration of polarization (40).

Comment 2: *In the supplementary data (p. S-3), it is listed that "the simulation length for each walker was 0.5 ns". To clarify, is the 0.5 ns the fixed time interval used for resampling in the weighted ensemble strategy? If so, I suggest referring to the 0.5 ns as the fixed time interval for resampling as "simulation length" may be confused with the length of the entire trajectory that consists of multiple fixed time intervals for resampling.*

Response: The text has been edited as recommended by the reviewer.

Revision: Page S-4, first paragraph.

The fixed time interval for resampling of each walker was 0.5 ns

Comment 3: *In the Methods, I assume that a Langevin thermostat was used since the simulations were carried out using the Amber software package. If this is the case, it is worth stating in the Methods that the simulations employed a stochastic thermostat and that this type of thermostat is*

required for the weighted ensemble strategy to generate continuous pathways with no bias in the dynamics.

Response: The reviewer is correct in that the Langevin thermostat was used, as stated on page S-5 in the paragraph **Molecular dynamics protocol**. To emphasize this, we have added a sentence in the paragraph Weighted ensemble MD simulations.

Revision: SI page S-4, first paragraph.

The WE simulations employed the Langevin thermostat, as a stochastic thermostat is required to for the WE strategy to generate continuous pathways with no bias in the dynamics.

Comment 4: *Depending on the number of transition events simulated using WE MD, it may be possible to estimate rate constants for transitions between the two binding modes of HID297 using the approach from Suarez et al. JCTC (2014). This calculation is not essential for the manuscript, but would provide additional valuable insights about the relevant timescales if there is sufficient data to perform the calculations.*

Response: We agree with the reviewer that an estimation of the rate constants for transitions between the two binding modes would give valuable insights. However, we do not have sufficient data at this point. We added a clarification in Concluding Discussion and cited the suggested reference.

Revision: Page 8, left column, second paragraph:

We also note that given converged WE simulations, the transition rate between the two binding modes may be estimated (43), which is a topic of future study.

Reviewer #4

Reviewer's remark: *The paper is very well written and nicely illustrated in the main text and also there is a lot of other informative data in the supplementary material. The employed methodologies (the weighted ensemble approach, continuous constant-pH MD with replica-exchange, and classical molecular dynamics) are valid and correctly used for establishing the binding modes and their stabilities of fentanyl in mOR.*

Response: We thank the reviewer for the favorable view of the paper and insightful comments. Below we give our response and revision.

Comment 1: *Although the paper is interesting it presents only one compound, fentanyl, and no other more potent derivatives, although it is claimed in the paper that "Our work provides a basis for understanding mOR activation by diverse fentanyl derivatives". The receptor activation is also not studied since the receptor structure is already activated.*

Response: We agree with the reviewer that the particular sentence in the Introduction might be misleading, as the paper only discusses fentanyl and its binding modes without considering the receptor conformational changes. That being said, what we meant is that our study provides a starting point for understanding how fentanyl activates mOR at a molecular level, as until now no crystal structure of mOR bound to fentanyl or analogs has been determined. A direct molecular dynamics simulation of the mOR activation by fentanyl is likely unfeasible, given the three orders of magnitude difference between the simulation and experimental time scale, e.g., the activation time of the class A GPCR α_{2A} adrenergic receptor was estimated as 40 ms (Vilardaga, Lohse et al., Nat.Biotechnol. 2003).

To address the reviewer's comment and clarify the above points, we revised the related text.

Revision: In Abstract:

Our work provides a starting point for understanding the molecular basis of mOR activation by

fentanyl which has many analogs emerging at a rapid pace.

Revision: Page 1, right column:

Our work provides a starting point for understanding how fentanyl activates mOR at a molecular level. Fentanyl analogs that can be significantly more potent and addictive than fentanyl are emerging on the dark market at a rapid pace.

Revision: Page 9, left column, first paragraph:

We also note that the present work is based on the activated structure of mOR and does not explore the large conformational changes of the receptor, which likely occur on a much slower timescale, e.g., the activation time of the class A GPCR α_{2A} adrenergic receptor was estimated as 40 ms (55). Notwithstanding the caveats, our detailed fentanyl-mOR interaction fingerprint analysis provides a basis for pharmacological investigations of fentanyl analogs, particularly how structural modifications alter the binding properties of newly identified fentanyl derivatives which may have increased potency and abuse potential.

Comment 2: *The binding site for BU72 is very big since the compound is big so the presented double binding mode of fentanyl may be a result of increased binding site of mOR-BU72. It would be good to make a long MD equilibration (about 1us) of empty receptor before docking of fentanyl.*

Response: The reviewer has raised an interesting point. In order to address this comment, we performed the binding pocket volume calculations using the crystal structure of the BU72-bound mOR and snapshots from the trajectory of the apo mOR simulation. After 110 ns, the volume stayed similar to the value from the crystal structure; however, after 500 ns, the volume increased from 359 ± 12 to 465 ± 51 Å³. Thus, prolonged apo simulation would not reduce the binding site volume. Note, we would not prolong the simulation even further, as without a bound agonist, the active structure might relax towards the inactive state. We have added the calculations and discussion in the SI.

Revision: SI, page S-6: A reviewer noted that BU72 is big and so the second binding mode may be the result of using the BU72-bound mOR crystal structure as a template to generate the initial structure for the fentanyl-mOR complex. The reviewer suggested running a 1- μ s MD equilibration of the empty receptor (with the goal to “shrink the binding pocket”). To address this comment, we performed the binding site volume calculations using POVME2.0 (S27, S28). All structures were first aligned using the binding site residues identified from the BU72-bound mOR crystal structure (PDB ID: 5C1M). All waters, ions, cholesterol, lipids, nanobody, and ligands were removed before the calculation. The binding pocket searching region is kept consistent throughout all systems by specifying an Inclusion box centered at the binding pocket center of mass (0,0,8) and sides of 12, 12, 15 Å in the x, y, and z direction, respectively.

The binding site volume based on the BU72-bound mOR crystal structure (with BU72 removed) is 371 Å³. After 110 ns of MD equilibration before docking fentanyl, the volume is 359 ± 12 Å³, which is similar to that from the crystal structure. To test if prolonged equilibration would shrink the binding site volume, we extended the simulation by 500 ns and found that the receptor’s cavity volume increased to 465 ± 51 Å³. An increase in volume can be rationalized as the result of relaxation and solvation of the binding site. Thus, a long MD equilibration of the empty receptor will not reduce the binding site volume, and the alternative binding mode of fentanyl is not a result of an expanded binding cavity due to the size of BU72.

We thank the reviewers again for their insightful comments, criticism, and suggestions. We have revised the manuscript to fully address and accommodate them. As a result, we believe the revised

manuscript is significantly improved and ready to be accepted.

Sincerely,

Jana Shen
Professor, Codirector,
Computer-Aided Drug Design
Center, American Chemical Society,
COMP Division Chair Elect 2021
Dept of Pharmaceutical Sciences
University of Maryland School of
Pharmacy, Baltimore, MD

Reviewer #1 (Remarks to the Author):

My majors concerns on the innovation of this work were totally ignored. The author only simply cited the already published two similar works and didn't clarified what exactly new and innovations in the cuurrent work.

I am pretty disappointed by this. Thus, I won't suggest for publication.

Reviewer #2 (Remarks to the Author):

The authors have, in my view, appropriately and sufficiently addressed all the reviewer comments. To reiterate from my previous report, the manuscript has multiple novel findings which could impact future research, including:

1. A better understanding of fentanyl binding to mOR, as well as the identification of a novel binding mode, which could influence our understanding of the binding of fentanyl analogs and other ligands to mOR.
2. An emphasis on the importance of investigating tautomerization states with regards to receptor/protein-ligand interactions.

One potential limitation is that, from this work alone, it is difficult to estimate how significant the alternative His297-binding mode of fentanyl is under physiological conditions. However, given the role of fentanyl in the opioid crisis and the possibility that this alternative binding mode could influence mOR-mediated responses to fentanyl, the findings in this work are meaningful. Additionally, this work will hopefully spur further research into the physiological prevalence and effects of the alternative fentanyl binding mode, as well as into the binding modes of fentanyl analogs and other mOR ligands.

In light of all these points, I fully support the acceptance of this manuscript.

Alissa M Hummer

Reviewer #4 (Remarks to the Author):

The Authors correctly and carefully replied to all the concerns and made the required additional MD simulation. The results confirm their initial findings therefore an existence of an additional binding mode of fentanyl presented in the paper is strengthened.

We are pleased to submit a revised manuscript entitled “How μ -Opioid Receptor Recognizes Fentanyl” by Quynh N. Vo, Paween Mahinthichaichan, Jana Shen, and Christopher R. Ellis. We would like this work to be reconsidered for publication as a regular article in *Nature Communications*. We thank the reviewers for reading our response and revision. Below we present our response to the reviewers’ new remarks.

Reviewer #1

Remarks to the Author: *My majors concerns on the innovation of this work were totally ignored. The author only simply cited the already published two similar works and didn’t clarified what exactly new and innovations in the cuurrent work.*

I am pretty disappointed by this. Thus, I won’t suggest for publication.

Response: We are very surprised by the remark of this reviewer. In our previous response letter, we explained in detail how our work differs from the previous two publications in both methodologies and findings. The differences are also explained in the manuscript. Furthermore, the first author of these previous publications (Dr. Lipinski) remarked on the novelty of our work and recommended publication, and so did the other three reviewers. Thus, we find the reviewer’s concern unjustified and we respectfully disagree with it.

Reviewer #2

Remarks to the Author: *The authors have, in my view, appropriately and sufficiently addressed all the reviewer comments. To reiterate from my previous report, the manuscript has multiple novel findings which could impact future research, including:*

1. A better understanding of fentanyl binding to mOR, as well as the identification of a novel binding mode, which could influence our understanding of the binding of fentanyl analogs and other ligands to mOR.

2. An emphasis on the importance of investigating tautomerization states with regards to receptor/protein-ligand interactions.

One potential limitation is that, from this work alone, it is difficult to estimate how significant the alternative His297-binding mode of fentanyl is under physiological conditions. However, given the role of fentanyl in the opioid crisis and the possibility that this alternative binding mode could influence mOR-mediated responses to fentanyl, the findings in this work are meaningful. Additionally, this work will hopefully spur further research into the physiological prevalence and effects of the alternative fentanyl binding mode, as well as into the binding modes of fentanyl analogs and other mOR ligands.

In light of all these points, I fully support the acceptance of this manuscript.

Response: We thank the reviewer for reading our response and revision. We also appreciate the reviewer’s summary and recommendation of publication.

Reviewer #4

Remarks to the Author: *The Authors correctly and carefully replied to all the concerns and made the required additional MD simulation. The results confirm their initial findings therefore an existence of an additional binding mode of fentanyl presented in the paper is strengthened.*

Response: We thank the reviewer for reading our response and revision.

In summary, we have made revision to fully accommodate the reviewers’ critiques and comments. We hope the manuscript is ready to be accepted for publication.